# Intranasal influenza-vectored COVID-19 vaccine restrains the SARS-CoV-2 inflammatory response in hamsters

Liang Zhang [1,8], Yao Jiang[1,8], Jinhang He[1,8], Junyu Chen[1,8], Ruoyao Qi[1,8], Lunzhi Yuan[1,8], Tiange Shao[2,8], Hui Zhao[3,8], Congjie Chen[1], Yaode Chen[1], Xijing Wang[1], Xing Lei[1], Qingxiang Gao [4], Chunlan Zhuang[1], Ming Zhou[1], Jian Ma[1], Wei Liu[1], Man Yang[1], Rao Fu[1], Yangtao Wu[1], Feng Chen[1], Hualong Xiong[1], Meifeng Nie[1], Yiyi Chen[1], Kun Wu[1], Mujin Fang[1,5], Yingbin Wang[1,5], Zizheng Zheng [1,5], Shoujie Huang[1,5], Shengxiang Ge [1,5], Shih Chin Cheng [4], Huachen Zhu [6,7], Tong Cheng [1,5], Quan Yuan [1,5], Ting Wu[1,5] ✉, Jun Zhang [1,5] ✉, Yixin Chen [1,5] ✉, Tianying Zhang [1,5] ✉, Changgui Li[3] ✉, Hai Qi [2] ✉, Yi Guan[6,7] ✉ & Ningshao Xia [1,5] ✉

The emergence of severe acute respiratory syndrome coronavirus-2 (SARS-CoV-2) variants and "anatomical escape" characteristics threaten the effectiveness of current coronavirus disease 2019 (COVID-19) vaccines. There is an urgent need to understand the immunological mechanism of broad-spectrum respiratory tract protection to guide broader vaccines development. Here we investigate immune responses induced by an NS1-deleted influenza virus vectored intranasal COVID-19 vaccine (dNS1-RBD) which provides broad-spectrum protection against SARS-CoV-2 variants in hamsters. Intranasal delivery of dNS1-RBD induces innate immunity, trained immunity and tissue-resident memory T cells covering the upper and lower respiratory tract. It restrains the inflammatory response by suppressing early phase viral load post SARS-CoV-2 challenge and attenuating pro-inflammatory cytokine (*Il6*, *Il1b*, and *Ifng*) levels, thereby reducing excess immune-induced tissue injury compared with the control group. By inducing local cellular immunity and trained immunity, intranasal delivery of NS1-deleted influenza virus vectored vaccine represents a broad-spectrum COVID-19 vaccine strategy to reduce disease burden.

The coronavirus disease (COVID-19) pandemic is a global disaster with unprecedented impact, causing serious public health hazards to human society[1]. SARS-CoV-2 (severe acute respiratory syndrome-related coronavirus) invades the host by binding to angiotensin-converting enzyme 2 (ACE2), a high-affinity receptor on respiratory epithelial cell surfaces. Currently, 50 vaccines have been approved for use, most of which are vaccinated by intramuscular injection. The

protective effects are mainly derived from the neutralizing antibody targeting spike antigen, and large-scale vaccination has effectively reduced SARS-CoV-2 symptomatic infection, hospitalization and death[2–4]. Antibody levels in the respiratory tract are 200-500 times lower than that in circulation[5], which leads to "anatomical escape" of SARS-CoV-2 in the upper respiratory tract since it is difficult to completely block the infection, especially after the peak vaccine-induced

A full list of affiliations appears at the end of the paper. ✉e-mail: wuting@xmu.edu.cn; zhangj@xmu.edu.cn; yxchen2008@xmu.edu.cn; zhangtianying@xmu.edu.cn; changguili@aliyun.com; qihai@tsinghua.edu.cn; yguan@hku.hk; nsxia@xmu.edu.cn

immune response period[6–8]. Furthermore, escape variants are emerging in an endless stream; for example, the Omicron mutant strain has the most significant changes in antigenicity and the immunity conferred after vaccinations and natural infection[9,10]. Finally, the virus is also hosted by several animal reservoirs such as minks, cats, deer's[11,12]. These realities portend that COVID-19 will coexist with humans for many years and will pose a continuing threat. Therefore, the development of broad-spectrum COVID-19 vaccines that rely on various immune mechanisms and different technical routes should be encouraged.

Theoretically, local protective immune factors in the respiratory tract should respond to SARS-CoV-2 infection in a timelier manner than effectors present in the peripheral lymph nodes and blood. Therefore, the development of COVID-19 vaccines via respiratory inoculation has become a hot pipeline shown positive effects in preclinical animal experiments, including vaccines based on adenovirus vectors, vesicular stomatitis virus vectors and other viral vectors[13–15]. An Ad-vectored trivalent COVID-19 vaccine expressing spike-1, nucleocapsid, and RNA-dependent RNA polymerase (RdRP) antigens from Afkhami et al. shows a good broad-spectrum protective effect against a variety of SARS-CoV-2 variant strains by intranasal vaccination[16]. In addition, trained immunity not dependent on specific antigen epitope also plays an important role in the broad-spectrum efficacy of this respiratory mucosal vaccine[16]. It would be difficult to achieve sterilizing immunity against SARS-CoV-2 infection by vaccination as the emerging variants and "anatomical escape" characteristics. Instead, inducing a local immune regulation mechanism that prevents an excessive inflammatory response in the respiratory tract will achieve broad-spectrum protection and reduce the COVID-19 disease burden. The broad-spectrum protective effect of NS1-impaired influenza virus on heterologous influenza virus challenge is independent of virus clearance and deserves attention[17]. Intranasal immunization with NS1-truncated virus (A/PR8/NS124) induces stronger effector T-cells and certain immunoregulatory mechanisms compared with wild-type H1N1 influenza strain (A/PR8/NSfull), which protects the organism against lethal heterologous A/Aichi/2/68 (H3N2) influenza virus challenge through significant attenuation of inflammation and pathology without inhibiting viral load[17]. These studies suggest raise a hypothesis that the vaccine-induced protective effect may be derived from immune regulation in the respiratory tract to prevent excessive inflammation, but not limited to block viral infection or suppress viral levels.

Modification of NS1 protein is a promising approach for the development of live-attenuated influenza viral vectors[18]. We previously developed an intranasal vaccine based on the NS1-deleted H1N1 vector carrying the gene encoding SARS-CoV-2-RBD (dNS1-RBD)[19,20]. A randomized placebo-controlled phase III clinical trial demonstrated that the vaccine has a good safety and broad-spectrum efficacy against Omicron symptomatic infection[21][ChiCTR2100051391], and was approved for emergency use in China on December 2, 2022, named Pneucolin. This vaccine prevents COVID-19 induced by Prototype, Beta and Omicron of SARS-CoV-2 challenge in hamster models in the absence of detectable neutralizing antibodies[19]. The immunological mechanisms that provide broad-spectrum protection remains unclear.

In this study, we investigated the protective immune response induced by dNS1-RBD. The broad-spectrum protective immunity induced by this vaccine mainly includes the following aspects: (i), innate immunity, various cytokines and chemokines containing anti-viral functions or initiating local immune responses were detected in lung tissue within 24 hours after vaccination; alveolar macrophages (AMs), dendritic cells (DCs), and natural killer (NK) cells were also activated; (ii), trained immunity that realizes the memory response of innate immunity by reprogramming the chromatin accessibility landscape which reshapes the immune response profile upon SARS-CoV-2 infection, with attenuation of pro-inflammatory factors and pathways;

iii), striking local T cell responses covering the upper and lower respiratory tract. Tissue-resident memory T cells (T_RMS) were detected in the nasal-associated lymphoid tissue (NALT) and the lung which supports the long-term protective effects. This intranasal vaccine represents an effective broad-spectrum COVID-19 vaccine strategy by inducing specific and non-specific protective immunity, particularly in the respiratory tract.

## Results

### Single-cell dynamics in lungs of dNS1-RBD and dNS1-Vector vaccinated mice

Bulk RNA-sequencing of lungs from vaccinated and uninfected animals were performed to map the dynamics of the vaccination-induced immune response (Fig S1A, B). Gene sets related to immune response, such as response to interferon-gamma, response to virus, and type I IFN signaling were enriched at 1-day post-immunization in the vaccinated mice (Fig S1C, D), reflecting the rapid recruitment and activation of immune cells upon vaccination. To define the kinetics of immune cell trafficking in greater detail and obtain a higher resolution of pulmonary immune responses triggered by this intranasally delivered vaccine, we performed scRNA-seq at multiple time points (Fig. 1A). After filtering out low-quality cells, red blood cells, and doublets, 158,844 cells were taken into further analysis.

Using canonical lineage-defining markers to annotate clusters, 14 major cell types in the lung were defined (Fig. 1B), including lymphoid, myeloid, and structural cells (Fig. 1C). The overall transcriptome features of dNS1-RBD and dNS1-Vector groups were comparable and greatly different from the wild-type CA04-infected and uninfected animals (Fig. 1D). The cell numbers and percentages of each cell type in each sample were calculated and compared across different treatments and timepoint. By 44 days post-infection (dpi.), vaccinated mice displayed a significantly high number of T cells, NK cells, and monocytes (mainly non-classical monocytes) compared to CA04-infected and uninfected mice (Fig. 1E). Previous studies have shown that the influenza A virus NS1 protein inhibits both innate and adaptive immune response, while truncation of NS1 protein elicits polyfunctional T-cell responses and increases the ability of the influenza A virus to stimulate innate immune responses[17,18]. Our previously published work showed that the key cytokine levels associated with the activation of innate immunity, such as IFN-α, IFN-γ, and TNF-α, etc. were rapidly upregulated 24 hours post-immunization in vaccinated mice lungs. Conversely, limited cytokine levels were detected in CA04 virus-infected mice at this time point but peaked later at 5 dpi., staying high with persistent lung inflammation[19]. These results suggested that the systemic response evoked by dNS1-RBD and dNS1-Vector is temporally and functionally distinct from its parental influenza virus. The deletion of the NS1 protein makes this influenza A virus-based intranasal vaccine better prime the host with improved immune efficacy.

### dNS1-RBD rapidly activates T and NK cells in the murine lung

In the cellular analysis of scRNA-seq data of lung samples from vaccinated, CA04 virus-infected, and uninfected mice, a significant increase of T and NK cells at 44 dpi. was observed (Fig. 1E). Flow cytometry analysis confirmed the significant increase in cell frequency and absolute number in vaccinated mice. T cells were clustered into 7 subpopulations based on their differentiation states, including effector, memory, and T_RMs, which possess a unique transcriptome profile different from central memory T cells and effector memory T cells (Fig. 2A–C and Fig S2A, B). When examining the functions of upregulated genes in pulmonary T cells of the dNS1-RBD and dNS1-Vector group, pathways related to cytokine-mediated signaling, cellular response to interferon-gamma, and defense response to virus were activated rapidly at 1 dpi., by contrast, the T-cell immune response triggered by wild-type CA04 virus appeared weak (Fig S2C). To further describe NK cell subpopulations, four distinct clusters were

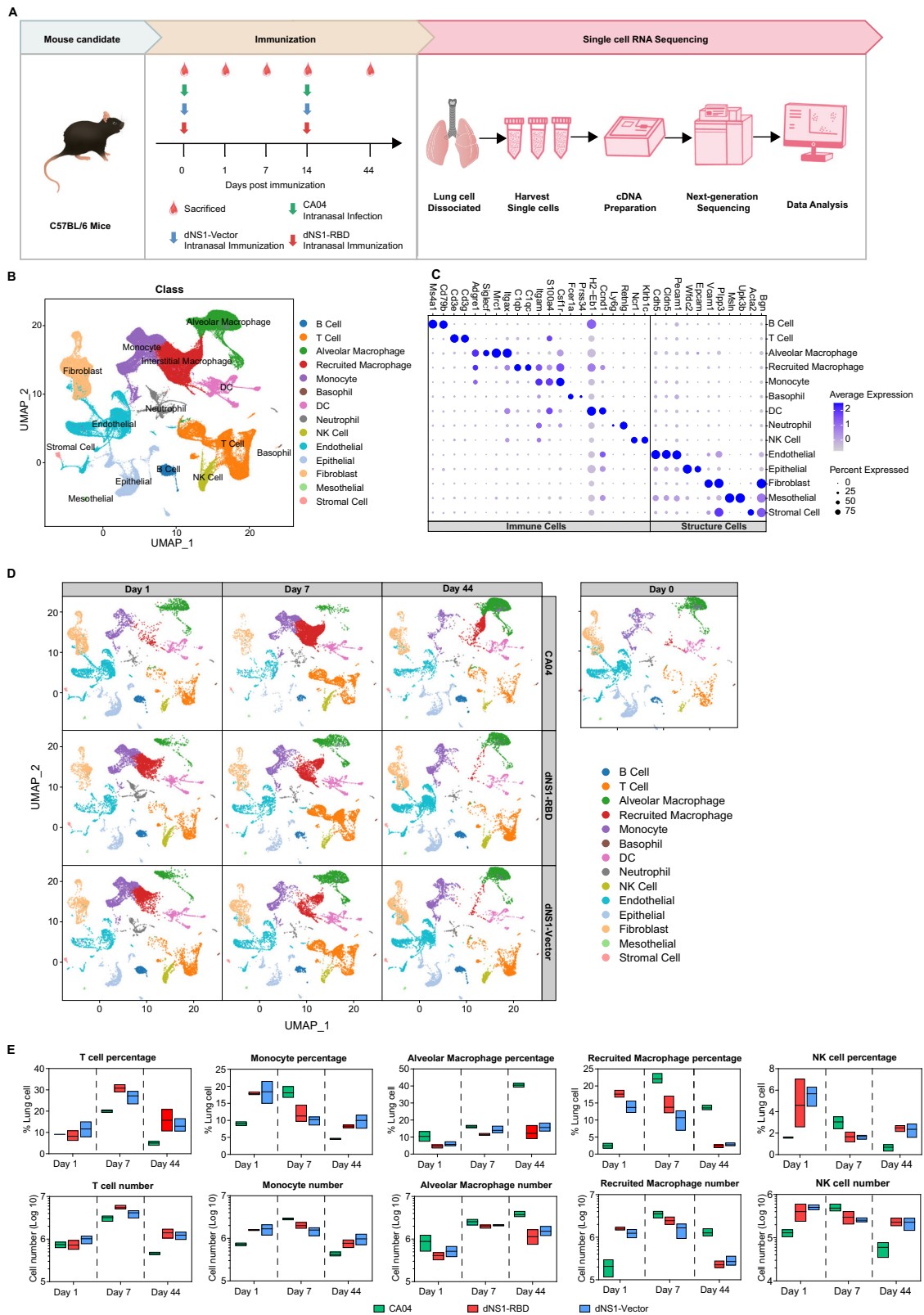

identified (Fig. 2D, F). In vaccinated mice, the fraction of NK cells expressing *Ifng* (cluster 1) expanded significantly at 1 dpi. and had declined to naïve levels at 7 dpi., by contrast, CA04 virus-infected mice maintained a low-level *Ifng*[+] NK cells response up to 7 days post-infection (Fig. 2E). By 44 dpi., vaccinated mice have a higher proportion (~75%) of NK cells expressing *Klrc2* (coding NKG2C protein) (cluster 2) as compared with CA04 virus-infected mice (~43%) and

uninfected mice (~66%) (Fig. 2E). Studies showed that a subset of *Klrc2*[+] NK cells might possess an antigen-unspecific memory-like feature with enhanced NK function[22,23].

To evaluate specific T cell response elicited by dNS1-RBD or dNS1-Vector, lung mononuclear cells were harvested at 1, 3, 5, and 14 days post single dose immunization. Intracellular IFN-γ expression was analyzed by flow cytometry after ex vivo stimulation using a 15-mer

**Fig. 1 | Cellular landscape of dNS1-RBD, dNS1-Vector and CA04 immunized identified by sc-RNA-seq. A** Flowchart showing an overview of the single-cell RNA-seq experimental design and schema, CA04 (green), dNS1-Vector (blue) and dNS1-RBD (red), created with adobe illustrator 2020. **B** Spectral UMAP plot of all cells analyzed from dNS1-RBD-immunized mice, dNS1-Vector-immunized mice, CA04-infected mice and control mice, showing identification of 14 different major cell types annotated by cell-type identity. **C** Dot plots of cells expressing selected canonical marker genes for identification of 14 different cell types in mice lungs across all samples. The size and color of the dot corresponding to the percentage of cells within the cell population expressing the gene and the average expression of

each marker genes, respectively. **D** UMAP plots showing lung cellular dynamics with dNS1-RBD-immunized, dNS1-Vector-immunized, and CA04-infected mice in different time points. **E** Box plots showing the proportion and cell numbers of T cells, monocytes, alveolar macrophages, recruited macrophages and NK cells across all groups. CA04 (green), dNS1-Vector (blue) and dNS1-RBD (red). $n = 2$–$3$ biologically independent mice/group. Box plots show the mean (center line) and minimum and maximum values (bounds of box). Source data are provided as a Source Data file. UMAP: Uniform Manifold Approximation and Projection; NK cells natural killer cells.

spike-peptide pool (Fig S2D). Single-dose immunization of dNS1-RBD elicits a strong IFN-γ-producing RBD-specific T-cell response in the lung (Fig. 2E, F). The spleens were also analyzed to compare local- and systemic antigen-specific CD8 T cell responses (Fig S2E); the strength of the immune response (number and percentage of total CD8a⁺ T cells) in spleen-specific T cells was weaker than that in local lung tissues (Fig S2F, G). A similar phenomenon was found in NK cells (Fig S2H–J). Enzyme-linked immunosorbent spot (ELISpot) assays of lymphocytes at 7 dpi, after ex vivo antigen-peptide stimulation revealed that dNS1-RBD induced a robust T cell response in the lung tissue by producing 1974/spot-forming cells (SFCs), which was 3.25 folds (606/SFCs) that of the spleen, and 16.18 folds (122/SFCs, $P < 0.0001$) that of the peripheral blood, respectively (Fig. 2G, H). By varying the vaccine dose to the animals, the spike-specific T cell responses were induced in a dose-dependent manner, and it was detectable even in the lowest dose group ($1 \times 10^2$ PFU/mL) (Fig. 2M).

### dNS1-RBD elicits tissue-resident memory T cell response in the upper and lower respiratory tract

Growing evidence supports a critical role for $T_{RMs}$ in coordinating effective defense against reinfection in the local tissue where they reside[24,25]. The nasal tissue of rodents contains NALT, which are mucosal-associated lymphoid organs embedded in the nasal passage[26]. A proportion of T cells within the murine NALT express a memory phenotype (CD44⁺), which is also observed in human tonsils (NALT equivalent)[27]. When mice were vaccinated with dNS1-RBD or dNS1-Vector, this memory phenotype population increased (Fig S3A, B). Memory CD8 T cells with the tissue-resident phenotype (CD69⁺ CD103⁺) also increased (Fig. 3A), both in number and proportion, 30 days after the second dose (Fig. 3B, C). These data suggested that nasal T cells may be induced to contribute to the protective immunity afforded by this vaccine. Similar increases in the lower respiratory tract were also observed (Fig. 3D, E, S3C-F). Similar to spike-specific T-cell responses, the tissue-resident memory T-cell responses also showed an obvious dose-response relationship (Fig. 3F, G).

　　Given the suboptimal local immunity in the respiratory tract induced by the current intramuscular (IM) route COVID-19 vaccine, potential strategies that would promote local immunity after IM immunization need to be identified. So a sequential immunization strategy containing StriFK, which is a SARS-CoV-2 subunit vaccine capable of inducing a potent humoral and cellular immune response in peripheral blood[28], and dNS1-RBD boosting were carried out to verify the local immunity enhancement brought by the boosting (Fig. 3H). Compared to mice that received IM immunization alone (group A), an intranasal (IN) booster vaccination of dNS1-RBD at 14 days after the last IM vaccination of StriFK (group B) led to a ~15-fold increase in lung resident T cell responses and promoted RBD-specific IFN-γ produced in the lung by ~45 fold (Fig. 3I, J). Mice in group B also showed increases in lung RBD-specific IFN-γ production and resident memory T cells as compared with mice that received dNS1-RBD alone (group C), although not statistically significant (Fig. 3I, J). These data suggest that by combining IM immunization with an intranasal dNS1-RBD boost, high levels of local lung immune responses were achieved. Previous studies have shown that patients surviving SARS infection generated

long-lasting memory T cells[29,30]. $T_{RMs}$ continue replenishment from circulating memory T cells and provide superior protection against local secondary infections[31]. These results highlighted the potential of $T_{RMs}$ as key targets for respiratory virus vaccines. Building up protection covering the upper and lower respiratory tract is optimal. Vaccination aimed at generating neutralizing antibodies alone may not be sufficient to protect against new and emerging variants of SARS-CoV-2. A more effective vaccine strategy should be taken into consideration for current and future pandemics.

### Macrophages and monocytes phenotypes in the murine lung after intranasal vaccination

In the early stages of infection at 1 dpi., robust, local transcriptome changes were observed in dNS1-RBD and dNS1-Vector immunized mice. Pathways of anti-viral innate immunity such as RIG-I-like receptor and Toll-like receptor are significantly activated. Responses to interferons were also coordinately upregulated, indicating that the synthesis and secretion of interferons are vigorous. As for the wild-type CA04 virus-infected mice, comparable mRNA expression change was observed at 7 dpi, reflecting a delayed immune response, since, at this time point, signal transduction pathways related to immune responses are negatively regulated in vaccinated mice (Fig. 4A). Viruses have evolved to escape the host's innate or non-specific immune system in the initial stage of infection. NS1-deleted H1N1-based intranasal vaccine ensures the rapid activation of the first line of defense and prevents subsequent inappropriate immune activation, which may result in severe lung pneumonia.

　　Lung macrophages, including alveolar macrophages and interstitial macrophages, play crucial roles in host defense, tissue repair, and organ homeostasis[32]. Sub-clustering of macrophages identified two major groups with unique features, tissue-resident alveolar macrophages (RAMs) and monocytes-derived recruited macrophages (RecMs) (Fig. 4B, C). Analysis of lung macrophages subclusters revealed that at the early stage of infection (1 dpi.), vaccinated mice lungs were particularly rich in recruited macrophages cluster 1 and cluster 3 (RecMs c01 and RecMs c03) (Fig. 4D, E). Of these, cluster 3 showed high expression of *Cd38*, *Cd86*, *Il1b*, *Nos2*, and *Cxcl10*, which have been related to M1 polarization[33–37], additionally, cluster 1 expressed high levels of *Isg15* and *Isg20* (Fig. 4C and Fig S4A). CA04-infected mice lungs were mainly represented by tissue-resident macrophages cluster 1 (RAMs c01) showing high expression of M2-like markers (*Mrc1*, *Ym1*, *Klf4*)[33–37] (Fig. 4C and Fig S4A). At this time point, lung macrophages represented a pro- and non-inflammatory functional state in vaccinated and wild-type virus-infected mice respectively. By 7 dpi., recruited macrophages have become the predominant macrophages population in CA04-infected mice (Fig. 4D, E and Fig S4A). Conversely, tissue-resident macrophages, predominantly cluster1 (RAMs c01), are the most abundant cells in dNS1-RBD and dNS1-Vector immunized mice (Fig. 4D, E and Fig S4A). Loss of alveolar macrophages (AMs) induced by infection could be replenished by self-renewal of surviving AMs or infiltration of monocyte-derived macrophages[38–40]. Pulmonary monocytes were divided into three subsets based on their surface expression of *Ly6c1*, *Ly6c2*, *Ccr2*, *Sell*, *Cx3cr1* (Fig. 4E, F)[41]. Murine Ly6C⁺ subsets, which are considered the

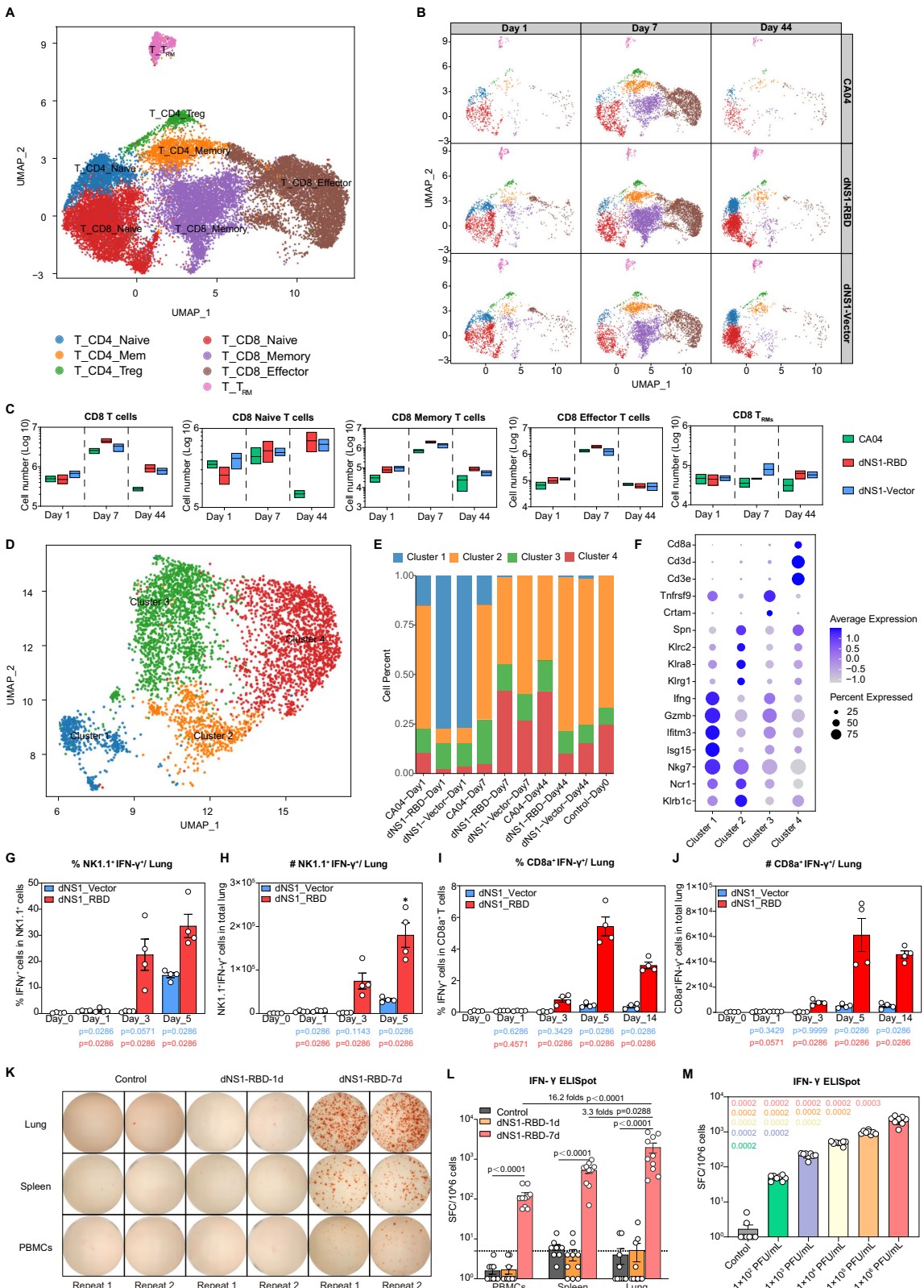

human counterpart of classical and intermediate monocytes, infiltrate into tissues, performing pro-inflammatory functions[42–44]. Ly6C⁻ monocytes, similar to human non-classical monocytes, enter into inflamed tissues to resolve inflammation and promote tissue repair. Under steady-state conditions, they exert a particular function of patrolling and surveillance, sensing viral nucleic acids, which is essential to initiate an innate immune response and restrict virus

infection[45–47]. As expected, vaccinated and CA04 virus-infected mice showed distinct temporal dynamics in pro-inflammatory monocytes recruitment (Fig. 4G, H). By 44 dpi., vaccinated mice have a higher proportion (~30%) of non-classical monocytes than that of CA04 virus-infected mice (~16%) and uninfected mice (~17%) (Fig. 4G). Innate immune cells react to viral infections but might also contribute to immunopathology. A harmonized balance between positive and

**Fig. 2 | Intranasal immunization of dNS1-RBD and dNS1-Vector activate T and NK cells. A** UMAP plot showing iterative clustering results of T cells reveals sub-populations corresponding to different states of differentiation. **B** UMAP plots showing T cells dynamics with dNS1-RBD-immunized, dNS1-Vector-immunized, and CA04-infected mice in different time points. **C** Box plots showing the cell numbers of all different CD8 T cells across all groups. CA04 (green), dNS1-Vector (blue) and dNS1-RBD (red). $n = 2–3$ biologically independent mice/group. Box plots show the mean (center line) and minimum and maximum values (bounds of box). **D** UMAP plot showing iterative clustering results of NK cells reveals subpopulations corresponding to different states of differentiation. **E** Stacked barplot showing the proportion of different NK cells. **F** Dot plots of cells expressing selected canonical marker genes for identification of different cell types in NK cells. **G, H** Statistical analysis plots for percentage (**G**) and cell number (**H**) of NK1.1$^+$ IFN-γ$^+$ in the lung. $n = 4$ biologically independent mice/group, dNS1-Vector (blue) and dNS1-RBD (red). **I, J** Statistical analysis plots for percentage (**I**) and cell number (**J**) of CD8a$^+$ IFN-γ$^+$ in the lung. $n = 4$ biologically independent mice/group, dNS1-Vector (blue) and dNS1-RBD (red). **K** Representative well images of the IFN-γ ELISpot response of the control group and dNS1-RBD group (1- and 7-days post-immunization). **L** SFCs per

million cells of IFN-γ from PBMCs, spleen, and lung were quantified after stimulation of a peptide pool covering the entire spike protein. $n = 10$ biologically independent mice/group, control (gray), dNS1-RBD-1d (orange) and dNS1-RBD-7d (red). **M** SFCs per million cells of IFN-γ from lung were quantified after stimulation of a peptide pool covering the entire spike protein in control and vaccinated group ($1 \times 10^6$ PFU/mL, $1 \times 10^5$ PFU/mL, $1 \times 10^4$ PFU/mL, $1 \times 10^3$ PFU/mL, $1 \times 10^2$ PFU/mL). $n = 8$ mice biologically independent mice/group. $P$-values are shown in the figure using different colors compared with the $1 \times 10^6$ PFU/mL group is represented in pink ($1 \times 10^5$ PFU/mL in orange, $1 \times 10^4$ PFU/mL group in yellow, $1 \times 10^3$ PFU/mL in purple, and $1 \times 10^2$ PFU/mL in green). Data are presented as mean ± SEM. Box plots show the mean (center line) and minimum and maximum values (bounds of box). Statistics analyses were Mann–Whitney tests (two-sided). Source data are provided as a Source Data file. $P$-values are shown in the figure using different colors in (**G–J**), the dNS1-Vector group compared with the control is represented in blue (dNS1-RBD in red). UMAP: Uniform Manifold Approximation and Projection; NK cells: natural killer cells; IFN-γ: interferon-gamma; SFCs: spot-forming cells; PBMCs: peripheral blood mononuclear cells; T$_{RMs}$: tissue-resident memory T cells.

negative regulation of innate immune responses is required to achieve the most favorable outcome for the host[48].

## dNS1-RBD immunization reprogrammed chromatin accessibility of alveolar macrophages and maintained the trained immunity phenotype

In recent years, the memory effect of the innate immune response (trained immunity) is thought to play an important role in broad-spectrum anti-infection immunity[49,50]. Innate immunity expresses a memory function that underlies broad heterologous immunity to antigenically diverse pathogens[51]. During trained immunity, the immunological memory of innate immune responses is broad and not limited to a single category of pathogens[51]. AMs are key players in the innate pulmonary defense during respiratory infection[52]. High MHC II expression is considered the trained phenotype of AMs[53]. In this study, MHC II expression on mouse AMs are significantly elevated after dNS1-RBD and dNS1-Vector vaccination (Fig. 5A). The upregulation of co-stimulatory molecules CD80 and CD86 was also observed (Fig. 5B, C). The trained immune phenotype of alveolar macrophages weakened as the vaccine dose decreased, although differences can still be observed compared to the control group (Fig. 5D). ATAC-seq analysis was used to identify the potential changes that may occur in the chromatin accessibility of AMs induced by intranasal immunization. Two months after the booster dose, KEGG and GO enrichment analyses showed that differential ATAC-peaks were significantly enriched in pathways related to innate immune response and involved Toll-like- and retinoic acid-inducible gene-1-like receptors (RLRs) (Fig. 5E). A total of 202 upregulated and 194 downregulated peaks were detected in the dNS1-RBD vaccinated mice (Fig. 5F). The dNS1-RBD group gained 810- and lost 760 lost open chromatin regions (OCRs) respectively, as compared with the control group (Fig. 5G). Peaks that were not present in the control group were induced in both the dNS1-RBD and dNS1-Vector groups in the regulatory region of MHC II genes (*H2-Aa and H2-Eb1*) (Fig. 5H). OCRs of several antiviral innate immune response genes including *TRIM25* and *IKBKB* are involved in RIG-I mediated antiviral signaling[54,55]. Toll-like receptors, *TLR3* and *TLR1*, were not detected in the control group compared to the dNS1-RBD group (Fig. 5H). AMs from dNS1-RBD hamsters gained and lost 913- and 1370 OCRs, respectively compared with the control at two weeks post-prime vaccination (Fig. 5I). The differential OCRs showed peaks in regulatory regions of pro-inflammatory cytokines, antiviral response, and toll-like receptor genes in both the dNS1-RBD and the dNS1-Vector group which was undetected in the control group (Fig. 5J).

These results indicated that the innate immune training might be induced by the intranasal vaccination of dNS1-RBD and dNS1-Vector. The observation of epigenetic programming of innate immune cell

supports the notion that the cross-protection effects of this intranasal vaccine might be partially mediated by trained immunity with enhanced non-specific effector responses.

## Intranasal immunization with dNS1-RBD vaccine protects hamsters from SARS-CoV-2 infection

Golden Syrian hamsters were challenged with beta SARS-CoV-2 variant by contact transmission to mimic patients with severe pneumonia caused by SARS-CoV-2[19]. Hamsters were sacrificed at 1-, 3-, 5 days dpi. for gross lung observation (Fig. 6A). Control hamsters showed continuous body weight loss beginning at 1 dpi. and exhibited weight loss of up to 11.97% at 5 dpi; in contrast, weight loss was not obvious in animals of dNS1-RBD group (mean: +1.24%) (Fig. 6B). Vaccinated hamsters showed a lower viral RNA load in nasal washings at 1 dpi. compared with the control hamsters. The significantly reduced viral RNA load (> 2.0 log) in nasal washings, trachea and lung at 1 dpi. (Fig. 6C), suggesting that an early innate immune response may be elicited in the respiratory tracts and thus restrict SARS-CoV-2 replication. A relatively lower viral RNA load was observed at 3 dpi. in trachea (> 1.0 log) and lung (> 2.0 log), which indicated that the intranasal vaccination inhibits the further infection of viruses from the upper to the lower respiratory tract of hamsters (Fig. 6C). The lung viral loads in vaccinated hamsters were slightly lower than that of the control hamsters at 5 dpi. although this was not statistically significant (Fig. 6C). Hematoxylin and eosin (H&E) staining of the lung lobes of SARS-CoV-2-infected hamsters in the dNS1-RBD group showed significant alleviation of the pathological changes. In contrast, control animals exhibited typical features of severe pneumonia including increased lung lobe consolidation and alveolar destruction, diffusive inflammation, hyaline membrane formation, and severe pulmonary hemorrhage (Fig. 6D). The apparent lesions in the dNS1-RBD group were markedly diminished, and no obvious viral-infection-related lung damage was observed in the gross lung images at 3- and 5 dpi. compared with the control group (Fig. 6D). The pathological severity scores of vaccinated hamsters were significantly lower than those of the control groups (Fig. 6C). The reduced lung inflammation could have been linked with the immediate anti-viral responses of hosts[56].

In order to validate the non-antigen-specific protective effect, hamsters were challenged by Beta variants through cohoused exposure after intranasally immunized with dNS1-RBD or dNS1-vector (Fig. 6E). The protective efficacy induced by the dNS1-vector was comparable to that induced by the dNS1-RBD at 1, 2, and 3 dpi. (Fig. 6F). Slightly enhanced protection was observed in hamsters vaccinated with dNS1-RBD, as reflected by their body weight change, with body weight change of +1.35% and -2.48% at 5 dpi. respectively (Fig. 6F). To assess the dose-effect relationship, a SARS-CoV-2

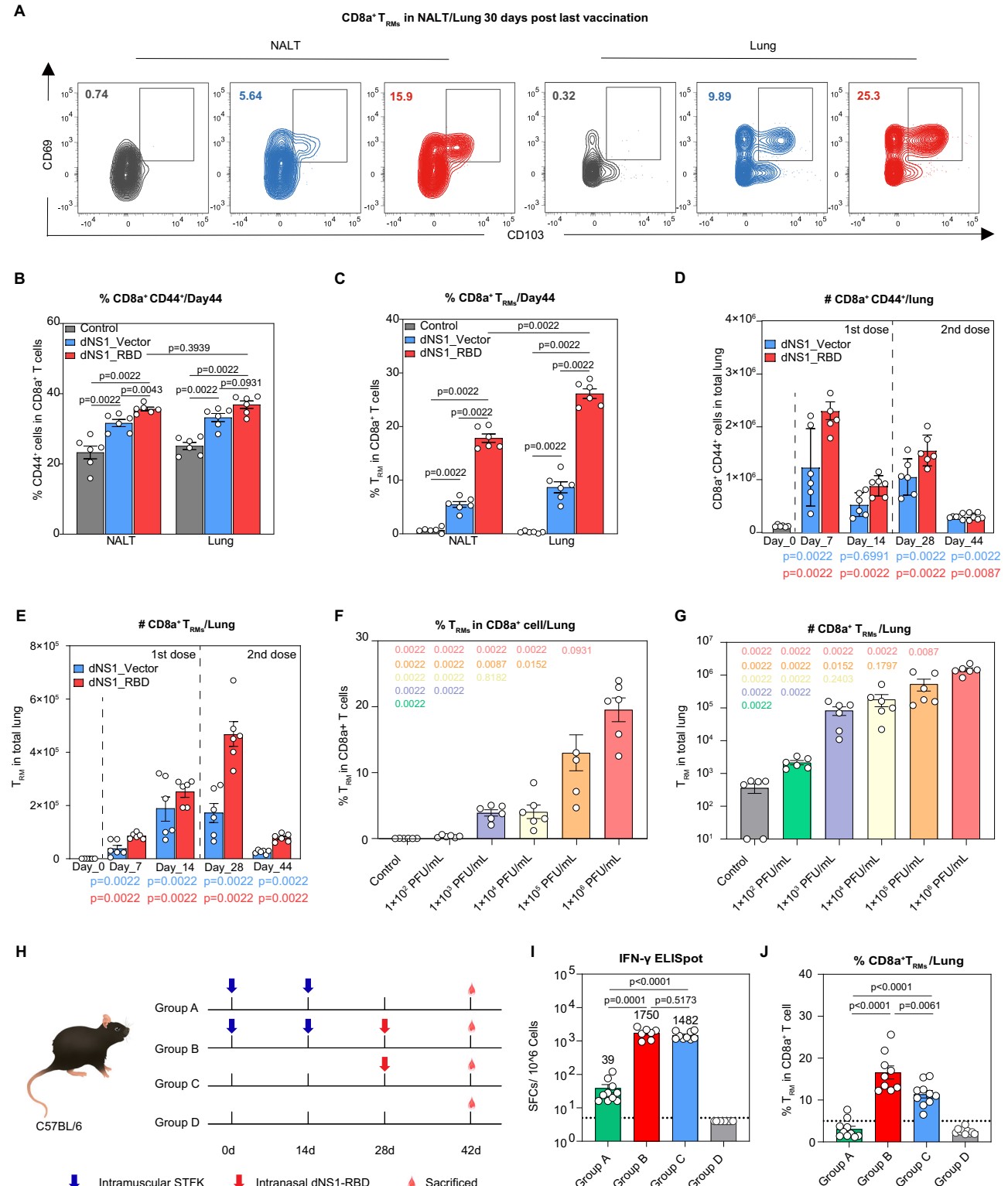

**Figure (A–J)**

A: CD8a⁺ T_RMs in NALT/Lung 30 days post last vaccination

challenge experiment was conducted in hamsters receiving varying doses of the vaccine. Consistent with the results of the cellular immune response, the results of body-weight change and lung tissue pathological damage showed that the low-dose groups ($1 \times 10^2$ PFU/mL and $1 \times 10^3$ PFU/mL) exhibited less protective effects than the higher-dose groups ($1 \times 10^5$ PFU/mL and $1 \times 10^6$ PFU/mL) (Fig S5A–C), and in which, the lowest-dose group that with detectable cellular immune responses still showed a protective effect compared to the control group.

**Intranasal immunization with dNS1-RBD vaccine prevents excessive inflammation triggered by SARS-CoV-2 infection**

Distinct gene expression signatures visualized by PCA from hamster lung samples (unchallenged and 1-, 3-, 5 dpi. of the unvaccinated control hamsters and dNS1-RBD vaccinated hamsters) following SARS-CoV-2 infection showed a tight clustering in each group. Unvaccinated control group samples at 3 dpi. and 5 dpi. were clearly separated in principal component 1. Meanwhile, all vaccinated group samples, unvaccinated samples at 1 dpi., and unchallenged samples were

**Fig. 3 | Intranasal immunized with dNS1-RBD induces T$_{RMs}$ cover respiratory tract. A** Representative flow cytometry contour plots for CD8a$^+$ T$_{RMs}$ in NALT and lung at 30 days post last vaccination, control (gray), dNS1-Vector (blue) and dNS1-RBD (red). **B**, **C** Bar graph showing the frequency of CD44$^+$ CD8a$^+$ T cells (**B**) and CD8+ T$_{RMs}$ (**C**) in NALT and lung 30 days post a prime-boost vaccination. $n = 6$ biologically independent mice/group, control (gray), dNS1-Vector (blue) and dNS1-RBD (red). **D**, **E** Bar graph depicting the absolute number of CD44$^+$ CD8a$^+$ T cells (**D**) and CD8a$^+$ T$_{RMs}$ (**E**) in the lungs at indicated time points after prime and booster immunization. $n = 6$ biologically independent mice/group, dNS1-Vector (blue) and dNS1-RBD (red). **F**, **G** Bar graph depicting the frequency (**F**) and the absolute number (**G**) of T$_{RMs}$ in the lungs after dNS1-RBD immunization ($1 \times 10^6$ PFU/mL, $1 \times 10^5$ PFU/mL, $1 \times 10^4$ PFU/mL, $1 \times 10^3$ PFU/mL, $1 \times 10^2$ PFU/mL). $n = 6$ biologically independent mice/group, compared with the $1 \times 10^6$ PFU/mL group is represented in pink ($1 \times 10^5$ PFU/mL in orange, $1 \times 10^4$ PFU/mL group in yellow, $1 \times 10^3$ PFU/mL in purple, and $1 \times 10^2$ PFU/mL in green). **H** Experimental design and schema, created with adobe illustrator 2020. **I** Bar graph depicting the numbers of IFN-γ SFCs from lung after stimulation of a peptide pool covering the entire spike protein. $n = 7$–10 biologically independent mice/group, group A: 2 doses of STFK (green), group B:2 doses of STFK and boost with dNS1-RBD (red), group C: single dose of dNS1-RBD (blue), group D: control (gray). **J** Bar graph depicting the frequency of T$_{RMs}$ in the lungs after booster immunization. $n = 9$–10 biologically independent mice/group, group A: 2 doses of STFK (green), group B:2 doses of STFK and boost with dNS1-RBD (red), group C: single dose of dNS1-RBD (blue), group D: control (gray). Data are presented as mean ± SEM. Statistics analysis were Mann–Whitney tests (two-sided). Source data are provided as a Source Data file. SFCs: spot-forming cells; T$_{RMs}$: tissue-resident memory T cells; NALT: Nasal-associated lymphoid tissue.

reflected in principal component 2 (6.8% of the variance) (Fig. 7A). Gene Ontology (GO) enrichment analysis of the genes upregulated in control hamsters were enriched for response to virus, cytokine production, and regulation of inflammatory response (Fig S6A). Genes involved in inflammatory cytokine production pathways such as *Il6, Il1b, and Ifng* were elevated at 1 dpi., peaked at 3 dpi., and remained high at 5 dpi. in control hamsters based on GO and KEGG pathway ssGSEA analysis (Fig. 7B, C). Excessive production of reactive oxygen species (ROS) resulting from apoptotic or inflamed cells may further aggravate inflammation[57] (Fig. 7C). Overactivated antiviral responses and excessive release of pro-inflammation cytokines are associated with severe immunopathology observed in unvaccinated hamsters. Notably, cytokine profiles for dNS1-RBD vaccinated hamsters were not significantly altered, demonstrating that they were protected from over-activated inflammation (Fig S6B). Dynamics of transcription levels in the vaccinated group exhibited a much steadier state, with mild elevation at 5 dpi. (Fig. 7C). The heatmap showed that different inflammatory signal pathways such as the IFN response, TNF signaling and their downstream signaling (for example, Jak-STAT and NF-κB signaling) were upregulated in the control group yet remained steady in the vaccinated group (Fig. 7C).

Dysregulation of cytokines and chemokines is closely associated with severe inflammation, leading to tissue damage and destruction. After SARS-CoV-2 challenge, the expression of *IFN-γ and CCL3* rapidly increased and remained high at 5 dpi. in the control group. Pro-inflammatory cytokines such as *IL-12, IL-6, IL-1B*, and *CXCL10* rapidly elevated after SARS-CoV-2 exposure in control hamsters. In addition, anti-inflammatory cytokines, such as *TGF-β* and *IL-10* also underwent a significant change. Conversely, in the vaccinated hamsters, most pro- and anti-inflammatory cytokines were relatively unperturbed in the initiation of viral infection and did not significantly progress in the later response, reflecting that homeostasis was maintained in the lung (Fig. 7D). During viral infection, the effective coordination of pro- and anti-inflammatory cytokines and their dynamic balance is required for the purpose of protecting the host. Distinct responses to SARS-CoV-2 infection between the dNS1-RBD vaccinated and unvaccinated hamsters were identified in transcriptome signatures. Prior vaccination with dNS1-RBD markedly alleviated the immunopathology caused by uncontrolled inflammatory responses. Approaches targeting single or multiple cytokines and pathways have been proposed for the abrogation of cytokine storms[58,59]. The potent immunomodulatory effects of dNS1-RBD facilitate it to simultaneously target multiple pro-inflammatory cytokines and pathways linked with severe COVID-19 pneumonia and poor outcomes in patients.

## Discussion

Although natural infections have occurred on a large scale and intramuscular COVID-19 vaccines have become widely available, controlling SARS-CoV-2 variants remains challenging. At the very beginning of the COVID-19 outbreak, our team began to develop an intranasal vaccine based on the NS1-deleted H1N1 vector carrying the gene encoding SARS-CoV-2-RBD (dNS1-RBD)[19]. To our knowledge, this is the first intranasal spray COVID-19 vaccine that was entered into a phase 3 clinical trial [ChiCTR2100051391]. In phases I and II clinical trials, the vaccine demonstrated an immunogenicity pattern that was highly consistent with animal studies-a weak peripheral immune response[20]. In recent phase III clinical trial results, this intranasal COVID-19 vaccine exhibited an encouraging efficacy combined with great safety profiles in the elderly or population with underlying chronic disease[21]. Thrillingly, this vaccine was approved for emergency use in China on December 2, 2022, named Pneucolin. This candidate intranasal spray influenza virus-vectored COVID-19 vaccine conferred broad-spectrum protection against symptomatic COVID-19 caused by SARS-CoV-2 Omicron variant in adults regardless of age or underlying medical conditions, both as the primary schedule or heterologous booster doses. dNS1-RBD conferred protection caused by SARS-CoV-2 infection without inducing significant neutralizing antibodies[19], and dNS1-vector also showed protective effects (Fig. 6E, F). The protective mechanism of this intranasal vaccine differs from the traditional vaccines and is vastly different from the previous understanding of vaccine immunity.

The protective immune mechanism induced by the vaccine includes at least the following four aspects: (1) cellular immune responses covering the upper and lower respiratory tract (Figs. 2, 3), (2) innate immunity (Fig. 4), (3) trained immunity (Fig. 5), (4) antibody targeting RBD[19].

RBD-specific cellular immune responses were induced in the lung, and the number of IFN-γ spot-forming cells per million lymphocytes was approximately 16 times that in peripheral blood. The response was generated on the 5$^{th}$ day after vaccination and persisted for at least six months in the periphery[19]. Earlier innate immune- and T-cell responses are of great significance for asymptomatic infection or mild disease after SARS-COV-2 infection[60–62]. T$_{RMs}$ in the respiratory tract and lungs are critical for controlling respiratory viral infections, and provide more timely, and stronger protective immunity than circulating T cells[24]. Meanwhile, since the upper respiratory tract is one of the first contact sites for inhaled pathogens, the T$_{RMs}$ in the nasal epithelium would prevent the transmission of virus from the upper respiratory tract to the lungs, thereby preventing the development of severe disease[26]. More importantly, combining intranasal vaccination of dNS1-RBD and traditional intramuscular vaccination could enhance the recruitment of tissue-resident memory T cells (Fig. 3F), providing durable and broad-spectrum immune protection[63–65]. Immunization of dNS1-RBD protects hamsters against challenge with SARS-CoV-2 in a dose-dependent manner, and its protective effect is consistent with the trend of cellular immune responses level. However, it is still difficult to establish a quantitative correlation between cellular response level and protection in this stage, which remains a challenge for the entire field.

There is now ample evidence that trained immunity is a component of host innate immune response to pathogens. Epigenetic modification of innate immune cells induced by vaccination with certain

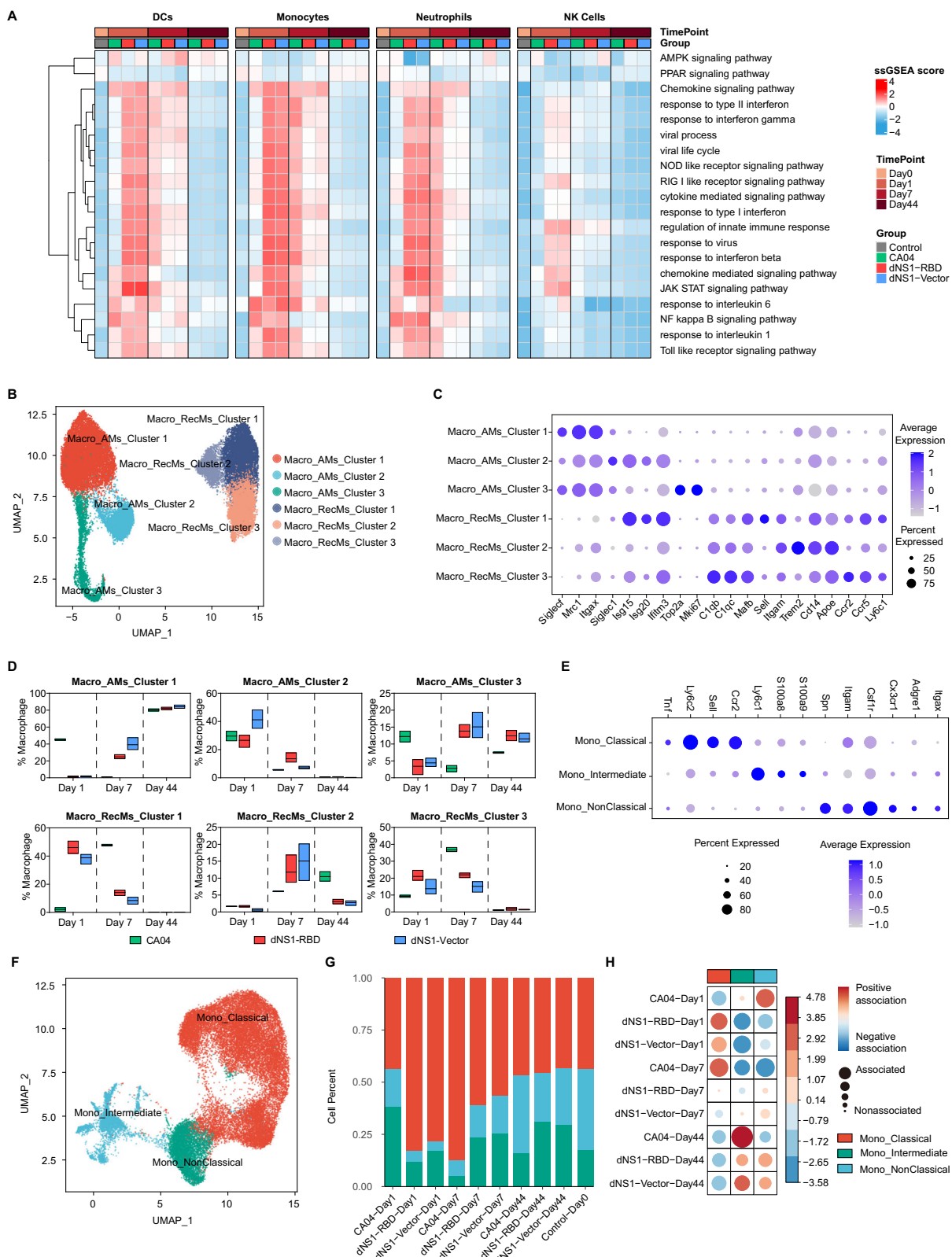

live vaccines could provide heterologous protection against unrelated pathogens within a certain period[49,50,66]. dNS1-RBD vaccination can also alter the host response pattern to SARS-CoV-2 by inducing trained immunity with (1) broad-spectrum antiviral and (2) anti-inflammatory effects. Vaccination of dNS1-RBD activated innate immune cells, accompanied by the alternations in chromatin accessibility, which might allow for a faster and stronger response to secondary infection

(Figs. 4, 5). In addition, the infected cells (including structural cells and immune cells) targeted by Influenza A H1 subtypes are highly overlapped with cells targeted by SARS-CoV-2, especially Omicron variant[67,68]. Immune regulation in structural cells is also widely reported[69]. Vaccination with dNS1-RBD may reshape the inflammation response by training of immune cells and structural cells, thus confer protection from homologous or heterologous virus infection.

**Fig. 4 | Anti-inflammation macrophages and monocytes were recruited in the lung. A** Heatmap showing the GO and KEGG pathway single-cell GSEA analysis of DC cells, monocytes, neutrophils and NK cells. **B** UMAP plot showing iterative clustering results of macrophages reveals subpopulations corresponding to different states of differentiation. **C** Dot plots of cells expressing selected canonical marker genes for identification of different cell types in macrophages. **D** Box plots showing the cell numbers of all different macrophage subtypes. Box plots show the mean (center line) and minimum and maximum values (bounds of box). CA04 (green), dNS1-Vector (blue) and dNS1-RBD (red). $n = 2$–$3$ biologically independent mice/group. **E** Dot plots of cells expressing selected canonical marker genes for identification of different cell types in monocytes. **F** UMAP plot showing iterative clustering results of monocytes reveal subpopulations corresponding to different states of differentiation. **G** Stacked barplot showing the proportion of different monocytes. **H** Dot plots of cell number of different cell types in monocytes. Correlation plot showing cluster enrichment in each group. Dot size proportional to Pearson's residual of the chi-squared test, while the color represents the degree of association from Pearson's chi-squared residuals. Source data are provided as a Source Data file. Box plots show the mean (center line) and minimum and maximum values (box). GO: Gene Ontology; KEGG: Kyoto Encyclopedia of Genes and Genomes; ssGSEA: single-cell gene-set enrichment analysis; DCs: Dendritic Cells; NK cells: natural killer cells; UMAP: Uniform Manifold Approximation and Projection; AMs: alveolar macrophages; RecMs: recruited macrophages.

Previous studies showed that cross-protective immune response could be induced across influenza viruses of A, B and C types because of common shared T cellular epitopes[70], however, the T cellular immune response induced by live-attenuated influenza vaccine is inadequate to provide broader cross-protection[71]. In a previous work researchers showed that mice immunized with the NS1-shortened H1N1 virus were better protected from lethal challenge with heterologous H3N2 virus as compared with wild-type influenza strain, and, in this case, no correlation was seen between increased survival and viral burden[17]. It is important to pay attention to the non-specific protection mechanisms related to the trained immunity. Emphasis should also be put on the immunoregulatory components of the immune system that may play an indispensable role in reducing the virus-triggered immunopathology.

Our findings also highlight the need for additional studies to quantitatively evaluate the contribution of each type of immunity induced by the vaccine to the efficacy against SARS-CoV-2 infection. So far, all of the vaccines that are successful against systemic respiratory viruses are systemically replicating live virus vaccines that fully encounter the host mucosal and systemic immune system[72] and that identifying strong immunologic correlates of protection against mucosal respiratory viruses in general is crucial in the development of next-generation vaccines. Previous studies have identified serum and mucosal immunoglobulin correlates and T cell immune correlates in humans after influenza infection[72]. However, a human influenza challenge study after vaccination with inactivated vaccines or live-attenuated influenza vaccine (LAIV), followed by LAIV challenge, was unable to find any immunologic correlates of protection[72] Therefore, additional immune correlate studies in humans are clearly needed and should be a research priority.

There are two main limitations in the study. First, while our study provides valuable insights into the efficacy of this vaccine, it is important to note that extrapolation of the induction of immunity by live vaccines in rodent animal models to humans has inevitable limitations. Second, we did not evaluate the potential impact of natural or acquired immunity on the nature of T-cell immunity induced by this vaccine against SARS-CoV-2.

Based on the findings of this study and previous knowledge, we suggest that the protective effects induced by intranasal vaccination could be not only dependent on viral clearance, but also through training the immune cells and structural cells in the respiratory tract, remodeling the immune microenvironment, and carrying out certain immunoregulatory effects after the heterologous challenge. This maintains the balance of the immune system and respiratory tissue and attenuates immune-induced tissue injury. Since intramuscular vaccines have been administered on a large worldwide, boosting with intranasal vaccines can establish more comprehensive immune protection in the "anatomical escape" location. At the same time, local cellular immunity and trained immunity in the respiratory tract are considered to be relatively broad-spectrum, which is beneficial for coping with the challenges caused by SARS-COV-2 variants. In conclusion, our findings contribute to the ongoing efforts to develop effective vaccines against SARS-CoV-2 and possibly other mucosal respiratory viruses.

## Methods

### Animal experiments

All animal experiments strictly followed the recommendations of the Guide for the Care and Use of Laboratory Animals. The animal studies were approved by the Institutional Animal Care and Use Committee (IACUC) of Xiamen University.

The hamster studies were performed in an animal biosafety level 3 (ABSL-3) laboratory (State Key Laboratory of Emerging Infectious Diseases, The University of Hong Kong).

C57BL/6 mice were purchased from Shanghai SLAC Laboratory Animal Co., Ltd. Golden Syrian hamsters were purchased from Beijing Vital River Laboratory Animal Technology Co., Ltd.

### Vaccine formulation

dNS1-RBD vaccine was prepared on a large scale at Beijing Wantai Biological Pharmacy Enterprise Co., Ltd., Beijing, China.

### Mouse immunization

Experimental animals were anesthetized with isoflurane, then intranasally immunized with 50 μL ($1 \times 10^6$ PFU/mL) of dNS1-RBD, whereas the control group was administered an equal volume of PBS. scRNA-seq analysis were performed on lung tissue collected from C57BL/6 mice after intranasal immunization. Tissues were collected on day 1 and 7 after single-dose vaccination, and on day 44 after booster vaccination.

Innate immune response analyses involved intranasal immunization of C57BL/6 mice (4 animals per group) with a single dose, and collection of pulmonary lymphocytes on day 1, 3, and 5 after vaccination.

ELISpot analyses of peripheral blood mononuclear cells (PBMCs), splenic lymphocytes, pulmonary lymphocytes, and lymph node cells involved intranasal immunization of C57BL/6 mice with a single dose (10 animals per group), and pulmonary lymphocytes were collected on day 1 and 7 after vaccination.

Intracellular cytokine staining (ICS) analyses of pulmonary lymphocytes involved intranasal immunization of C57BL/6 mice with a single dose (four animals per group), and collection of pulmonary lymphocytes on day 1, 3, 5, and 14 after vaccination.

Tissue-resident T-cell analyses involved intranasal immunization of C57BL/6 mice with a single dose (6 animals per group). Pulmonary lymphocytes were collected on day 7 and 14 after vaccination. In another experiment, mice were intranasally immunized with a booster dose on day 14 and pulmonary lymphocytes were harvested on day 14 and 30 after boost vaccination. For quantitative assessment of the vaccine-induced responses, pulmonary lymphocytes were harvested on day 14 after a booster-dose immunization with varying doses of dNS1-RBD, including $1 \times 10^6$ PFU/mL, $1 \times 10^5$ PFU/mL, $1 \times 10^4$ PFU/mL, $1 \times 10^3$ PFU/mL, $1 \times 10^2$ PFU/mL.

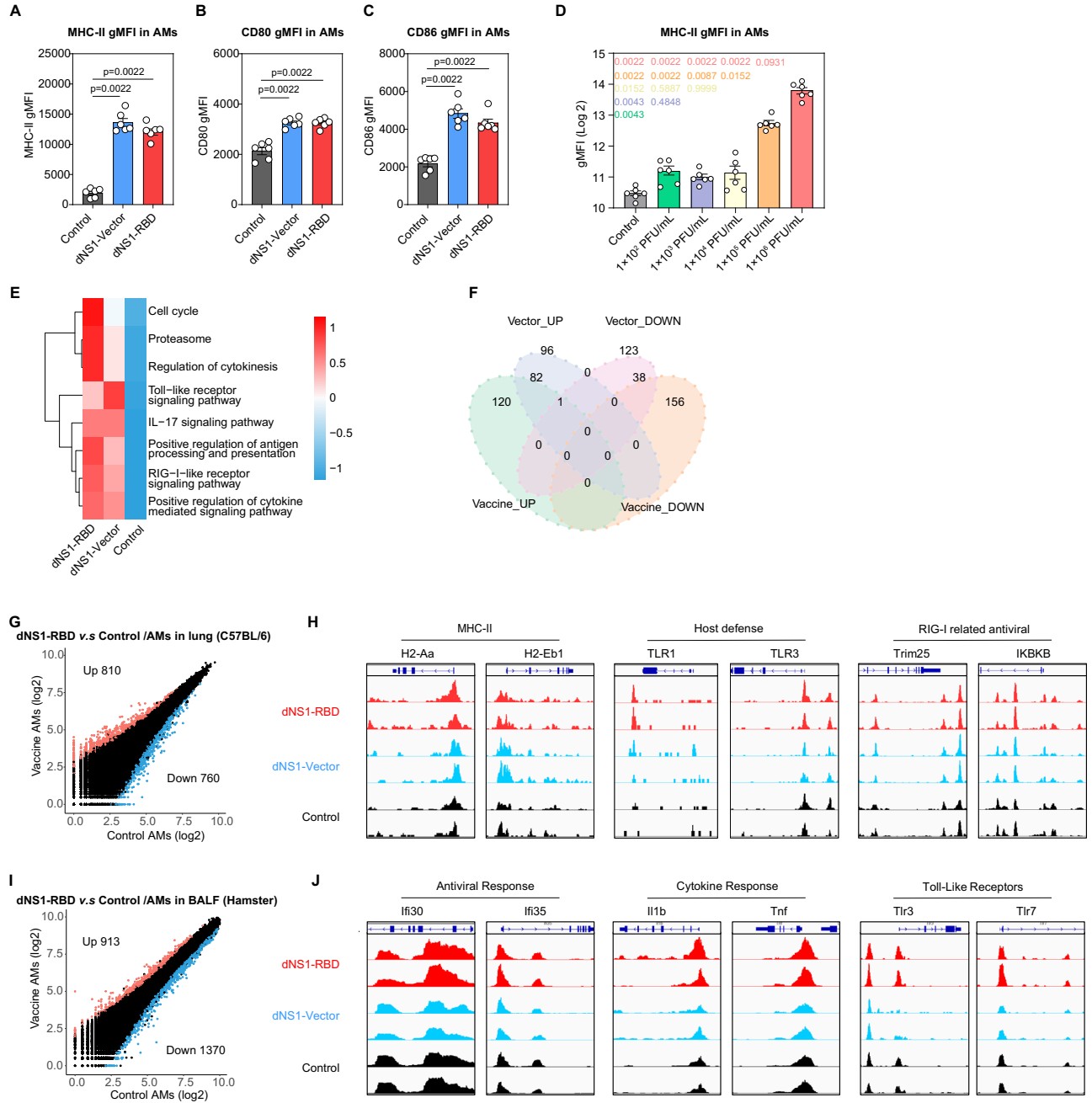

**Fig. 5 | Trained phenotype of alveolar macrophages induced by dNS1-RBD and dNS1-Vector.** **A**−**C** Statistical analysis plots for gMFI of MHC II (**A**), CD80 (**B**) and CD86 (**C**) on AMs in C57BL/6 mice after immunization (dNS1-RBD or dNS1-Vector). $n = 6$ biologically independent mice/group, control (gray), dNS1-Vector (blue) and dNS1-RBD (red). **D** Statistical analysis plots for gMFI of MHC II on AMs in C57BL/6 mice after vaccine immunization ($1 \times 10^6$ PFU/mL, $1 \times 10^5$ PFU/mL, $1 \times 10^4$ PFU/mL, $1 \times 10^3$ PFU/mL, $1 \times 10^2$ PFU/mL). $n = 6$ biologically independent mice/group, compared with the $1 \times 10^6$ PFU/mL group is represented in pink ($1 \times 10^5$ PFU/mL in orange, $1 \times 10^4$ PFU/mL group in yellow, $1 \times 10^3$ PFU/mL in purple, and $1 \times 10^2$ PFU/mL in green). **E** KEGG and GO enrichment result of shared peaks, with color bar on $-\log_{10}$ ($q$-value) scale. **F** Venn diagram showing differential ATAC-seq peaks (annotated as promoters) for dNS1-RBD or dNS1-Vector compared to the Control group. **G** Scatter-plot of differentially detected ATAC-seq peaks ($\log_2 FC > 1.5$, $q$-value < 0.05) of AMs (Lung) in the dNS1-RBD vaccination group compared to the control group in C57BL/6. **H** IGV tracks showing

differentially detected peaks related to host defense and antiviral response in C57BL/6 mice from dNS1-RBD vaccination group, dNS1-Vector vaccination group and control group. $n = 2$ biologically independent mice/group, control (black), dNS1-Vector (blue) and dNS1-RBD (red). **I** Scatter-plot of differentially detected ATAC-seq peaks ($\log_2 FC > 1.5$, $q$-value < 0.05) of AMs (BALF) in dNS1-RBD vaccination group compared to control group in hamsters. **J** IGV tracks showing differentially detected peaks related to host defense and antiviral response in hamsters. $n = 2$ biologically independent hamsters/group, control (black), dNS1-Vector (blue) and dNS1-RBD (red). Data are presented as mean ± SEM. Statistics analyses were Mann−Whitney tests (two-sided). Source data are provided as a Source Data file. gMFI: Geometry Mean Fluorescence Intensity; MHC-II: major histocompatibility-II; AMs: alveolar macrophages; GO: Gene Ontology; KEGG: Kyoto Encyclopedia of Genes and Genomes; ATAC-seq: Assay for Targeting Accessible-Chromatin with high-throughput sequencing; IGV: Integrative genomics viewer; BALF: bronchoalveolar lavage fluid.

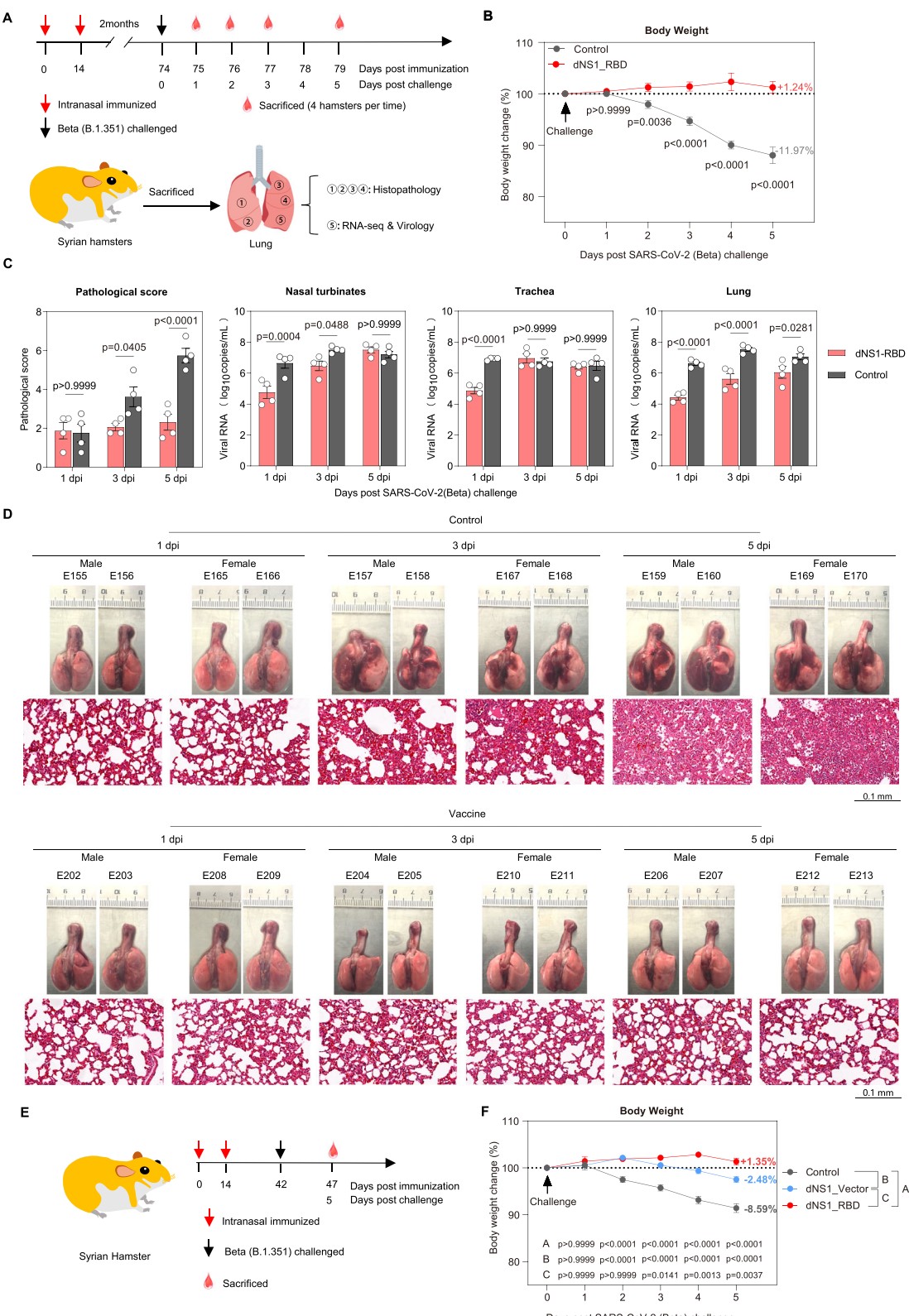

## Hamster immunization and infection

The experimental hamsters (male: female = 1:1) were anesthetized with isoflurane and intranasally immunized with 100 µL of the vaccine or vector ($1 \times 10^6$ PFU/mL, 0/14 day), whereas the control group was administered with an equal volume of PBS. Hamsters were further evaluated by direct contact challenge with SARS-CoV-2. The donor hamsters (carrying the virus) were intranasally infected with $1 \times 10^3$ PFU

of SARS-CoV-2. Each donor was transferred to a new cage and co-housed with the hamster of the dNS1-RBD group or control group for one day. The donor was then isolated, and the other hamsters were observed. Weight changes and typical symptoms (piloerection, hunched back, and abdominal respiration) were recorded daily after virus inoculation or contact. Hamsters were euthanized for tissue pathological and virological analyses and RNA-seq on day 1, 3, and 5 after

**Fig. 6 | Intranasal immunization with dNS1-RBD vaccine protects hamsters from SARS-CoV-2 infection. A** Schema of the experimental design. On days 1, 3, and 5 after cohoused exposure, hamsters from vaccinated and control groups were euthanized for analyses. $n = 4$ biologically independent hamsters/group, created with adobe illustrator 2020. **B** Body weight changes of hamsters after cohoused exposure were plotted. The average weight loss of each group at 5 dpi. is indicated as a colored number. $n = 4$ biologically independent hamsters/group, control (gray) and dNS1-RBD (red). **C** Bar graph showing the pathological severity scores of lungs and the viral RNA loads from nasal turbinate, trachea, and lung. $n = 4$ biologically independent hamsters/group, control (gray) and dNS1-RBD (red). **D** Gross lung

images and H&E-stained lung sections from dNS1-RBD vaccinated and control groups. Experiments were repeated 3 times independently with similar results. **E** Schema of the experimental design, created with adobe illustrator 2020. Hamsters were intranasally vaccinated with dNS1-RBD or dNS1-Vector. **F** Body weight changes of hamsters after cohoused exposure were plotted. The average weight loss of each group at 5 dpi. is indicated as a colored number, $n = 8$ biologically independent hamsters/group, control (gray), dNS1-Vector (blue) and dNS1-RBD (red). Data are presented as mean ± SEM. Statistical analysis was two-way ANOVA with Bonferroni's multiple comparisons test. Source data are provided as a Source Data file.

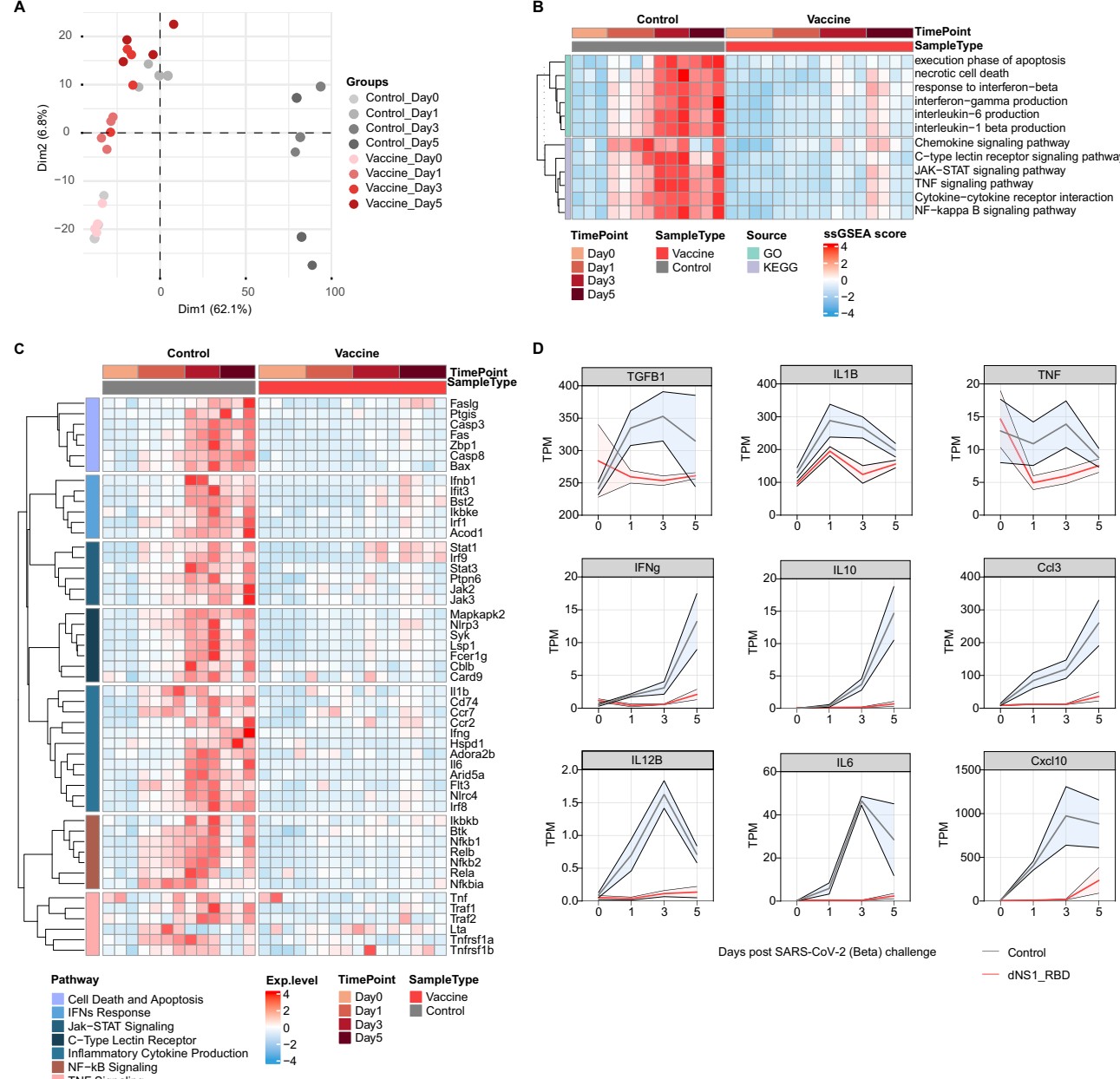

**Fig. 7 | Transcriptome analysis of lung reveals distinct immune response between dNS1-RBD immunized and control hamster. A** PCA Plot showing the global differences between vaccinated and control groups. **B** Heatmap showing the GO and KEGG pathway ssGSEA enrichment analysis results. **C** Heatmap visualization of scaled gene expression levels (TPM) for selected

pathways of interest. **D** Dynamic expression of cytokines in the lungs of hamsters, with error bars shaded (using standard error of the mean), dNS1-RBD (red line), control (gray line). PCA: Principal component analysis; GO: Gene Ontology; KEGG: Kyoto Encyclopedia of Genes and Genomes; ssGSEA: single-cell gene-set enrichment analysis.

challenge. In another experiment, all vaccinated (dNS1-RBD and dNS1-Vector) and control hamster were euthanized at 5 days post-infection. For dose-effect relationship assessment, hamsters were euthanized for tissue pathological and virological analyses at 5- and 7-days post-infection. Virus challenge studies were performed in an animal biosafety level 3 (ABSL-3) facility. The SARS-CoV-2 strain used in this study was B.1.351 variant AP100 (hCoV-19/China/AP100/2021; GISAID accession No. EPI_ISL_2779638)

### Organ-specific sample collection and organ dissociation

Mice were euthanized by exsanguination, and IACUC guidance was approved. Mice were transferred to a biosafety cabinet and their organs were carefully separated. All cells were counted using Count-Star software.

### Lung

Lungs were cut into 0.5-cm pieces, placed in gentleMACS C tubes (Miltenyi) containing collagenase type IV (Gibco) and DNase I (Roche) in PBS containing 2% FBS, and dissociated using a gentleMACS Dissociator (Miltenyi; program m_lung_01). A single-cell suspension was obtained by digesting tissue through a 70 μm cell strainer, and centrifugation at 300 g for 5 min at 4 °C. After centrifugation, 1 mL of cold red blood cell lysis buffer (Solarbio) was added for 2 min to lyse red blood cells. The reaction was stopped by adding 10 mL of cold PBS containing 2% FBS and washed once to remove residual red blood cell lysis buffer. Lymphocytes were obtained from the resulting cell suspensions using density gradient centrifugation (Percoll, SIGMA-ALDRICH). Cells were recovered at the interface of the 80% Percoll layer and the 40% Percoll layer, then washed with PBS + 2% FBS at 500 g for 5 min to remove excess Percoll.

### Lymph nodes

Cervical lymph nodes were carefully pinched with tweezers and rinsed several times with cold PBS containing 2% FBS. Lymph nodes were ground and passed through a 70 μm cell strainer. Lymphocytes were washed once and resuspended in PBS containing 2% FBS.

### Spleen

Mice were euthanized and their spleen were carefully separated and rinsed several times with cold PBS containing 2% FBS. Spleen were ground and passed through a 70 μm cell strainer and the cells were centrifuged at 300 g for 5 min at 4 °C. 10 ml of cold red blood cell lysis buffer (Solarbio) was added, and the samples were incubated for 5 min at 4 °C. The reaction was stopped by adding 20 ml of cold PBS containing 2% FBS and washed once to remove the residual buffer. Lymphocytes were washed once and resuspended in PBS containing 2% FBS.

### PBMCs

Mouse peripheral blood was transferred into a centrifuge tube containing sodium heparin, then 4 mL PBS buffer was added and transferred to SepMate™ PBMC isolation tubes (STEMCELL). PBMCs used density gradient centrifugation at 1200 g for 10 min at 25 °C (Ficoll-Paque PREMIUM, GE). PBMCs are obtained from the middle layer cells.

### Nasal-associated lymphoid tissue (NALT)

The lower jaw of the mouse was removed, and a surgical knife was used to carefully cut and excise the upper palate by following the inner contour of mouse incisors and molar teeth. The tissue was digested at 37 °C with collagenase type IV (Gibco) and DNase I (Roche) in PBS containing 2% FBS. A single-cell suspension was obtained from the digested tissue using a 70 μm cell strainer.

### Flow cytometry

The expression of phenotypic markers, activation markers, and cytokines was evaluated. Briefly, cells were washed and blocked with antiCD16/CD32 (clone 2.4G2) in 0.5 % FBS-PBS for 30 min on ice, then stained with fluorochrome-labeled mAbs for 30 min on ice. ICS assays involved stimulating each sample with pooled spike peptides (1.0 μg/mL) in a U-bottom plate and incubating at 37 °C for 18 h. Golgi-Plug (BD Biosciences) was added to the culture at a final concentration of 1:1,000 and cells were further incubated for 6 additional hours. After incubation, cells were washed and stained with fluorochrome-labeled mAbs for 30 min on ice. The stained cells were fixed and permeabilized with BD Cytofix/Cytoperm (BD Biosciences, San Jose, CA, United States) according to the manufacturer's instructions. The cells were washed and intracellularly stained with fluorochrome-labeled mAbs for 45 min on ice. The antibody reagents used in this study include: CD4 [Clone GK1.5, FTIC], CD8a [Clone 53-6.7, PE/Cy7], NK1.1 [Clone PK136, PerCP/Cy5.5], CD64 [Clone X54-5/7.1, PE /Cy7], CD170 [Clone S17007L, PE], CD11b [Clone M1/70, FITC], CD86 [Clone PO3, BV605], CD11c [Clone N418, BV421], CD45.2 [Clone 104, APC/Cy7], CD4 [Clone GK1.5, APC], CD8a [Clone 53-6.7, FITC], CD103 [Clone 2E7, PE], CD69 [Clone H1.2F3, BV421], CD44 [Clone IM7, PE/Cy7], CD45.2 [Clone 104, PerCP/Cy5.5], CD80 [Clone 16-10A1, PE /Cy7], CD11b [Clone M1/70, PE], CD317 [Clone 927, BV421], Ly-6C [Clone HK1.4, APC/Cy7], MHC class II [clone M5/114.15.2, APC], CD11c [Clone N418, APC]), cytokine expression (IFN-γ [clone XMG1.2, APC]), and a LIVE/DEAD® Fixable Aqua Dead cell stain kit was also used. Stained cells were processed using a BD LSRFORTESSA X-20 (BD Biosciences) flow cytometry system according to the manufacturer's instructions. Data were analyzed using FlowJo X 10.0.7r2 and GraphPad Prism 8.

### ELISpot assay

Dissociated PBMCs, splenic lymphocytes, pulmonary lymphocytes, and lymph node cells were plated at $2.5 \times 10^5$ into each well of a mouse IFN-γ ELISpot plate (Dakewe Biotech). And for dose-response relationship, $2 \times 10^5$ were plated. Samples were stimulated using pooled Spike peptides of SARS-CoV-2 (Final concentration:1 μg/mL, 15-mer peptide with 11 amino acids covering the spike region, Genscript) and cultured at 37 °C with 5% CO₂ for 20 h. Spots were scanned, counted, and quantified using the CTL S6 Universal Analyzer (Cellular Technology Limited) according to the manufacturer's instructions.

### SARS-CoV-2 and dNS1-RBD RNA quantification

Detection of viral RNA levels was performed in hamster lungs using quantitative RT-PCR. Lung tissue was homogenized using TissueLyser II (Qiagen, Hilden, Germany) and RNA extraction was performed according to the manufacturer's instructions (QIAamp Viral RNA mini kit, Qiagen). Viral RNA quantification was performed by measuring the copy number of the N gene using a SARS-CoV-2 RT-PCR kit (Wantai, Beijing, China), whereas CA4-dNS1-nCoV-RBD was quantified using RBD-targeted primers and the NS gene.

### Histopathology

Hamster lung tissues were fixed with 10% formalin for 48 h, embedded in paraffin, sectioned, and subjected to hematoxylin and eosin (H&E) staining. Whole-slide images of the lung sections were captured using a Leica Aperio Versa 200 microscope. Pathological lung lesions were scored based on (i) Alveolar septum thickening and consolidation; (ii) hemorrhage, exudation, pulmonary edema, and mucous; (iii) recruitment and infiltration of inflammatory immune cells. For each lobe, a score was determined based on the severity and percentage of injured areas. Four independent lobes of the lung tissues were scored and average lung pathological score of each individual hamster was used for pathological evaluation.

### Single-cell RNA sequencing

Single-cell suspensions with $1 \times 10^5$ cells/mL in concentration in PBS were prepared. Single-cell suspensions were loaded into microfluidic devices and scRNA-seq libraries were constructed following the

Singleron GEXSCOPER protocol by GEXSCOPER Single-Cell RNA Library Kit (Singleron Biotechnologies). Indivaidual libraries of each sample were then dilyted to 4 nM, pooled and sequenced on Illumina novaseq 6000 with 150 bp paired end reads.

## Analysis of single-cell RNA sequence
**Pre-process.** The pre-process of the scRNA-seq data was mainly based on the CeleScope software (version 1.10.0) under a CentOS 7.9 platform. The mouse genome (GRCm38) fasta file and gene annotation files were downloaded from ENSEMBL(ftp.ensembl.org), and then the index was built by using 'celescope rna mkref' function. The raw data in fastq format were processed and mapped to the mouse genome (GRCm38) by following the shell scripts generated with the celescope multi_rna pipeline with '--cell_calling_method auto --allowNoPolyT --mod shell' command, which generated the gene expression matrix of each sample for further analysis.

## scRNA-seq quantifications, clustering, and reduction
R package Seurat (version 4.1.1) was used for single-cell data analysis under an R 4.1.3 platform. The original expression matrix for each sample was imported and filtered with at least 300 genes present in each cell and at least 3 cells per gene expressed. Furthermore, cells with high detection rates of mitochondrial gene expression were removed. Then the raw expression matrix was logged normalized and 6000 highly variable genes were scaled for the dimension reduction. Principal component analysis (PCA) was calculated based on the selected variable genes, and the most significant 50 principal components (PCs) were chosen as the input data for Uniform Manifold Approximation and Projection (UMAP) for visualization and identifying clusters of cells by a shared nearest neighbor (SNN) modularity optimization based on Louvain methods.

## Removing doublet, cell type annotation, and cell cycle detection
To detect the doublet, DoubletFinder (version 2.0.3) were performed on each sample's dataset, which cells were annotated as doublet were removed, furthermore those cells express conserved marker genes from two cell types were also considered as doublet and removed artificially. And to correct the batch effect and classify the cell types within the dataset, we redo the clustering and reduction with the Reciprocal PCA (RPCA) integration method. Then FindAllMarkers function were used to find the marker genes for each cluster and referring to these unbiased marker genes and conserved marker genes based on prior knowledge, all clusters were classified into specific cell types. Cell cycle analysis were performed through the CellCycleScoring pipeline within the Seurat Package.

## Single-cell differential expression gene analysis and gene-set enrichment analysis
The differential expression gene analysis was performed by using the MAST methods (version 1.20.0) provided in the R package Libra (version 1.0.0). In each comparison, all the genes were ranked by log fold change, and then the GSEA enrichment analysis were performed by using R package clusterProfiler (version 4.2.2) and fgsea (version 1.20.0).

## Single-cell ssGSEA analysis
The ssGSEA analyses were performed on the raw count matrix by using R packages GSVA (version 1.42.0), GSEABase (version 1.56.0), and Misgdbr (version 7.5.1).

## Bulk RNA sequencing
The hamster lung lobe was removed, shredded into small pieces and stored in RNA Later Solution (Thermo Fisher Scientific) for a maximum of 24 h at 4 °C. Lung tissue was homogenized using TissueLyser II (Qiagen, Hilden, Germany) and RNA extraction was performed according to the manufacturer's instructions (QIAamp Viral RNA Mini Kit (Qiagen)). The RNA samples were sent to OE Biotech Co., Ltd. (Shanghai, China) for RNA purification, cDNA library construction, and sequencing.

## ATAC sequencing
Hamster lungs were collected 2 months post-vaccination and sorted for bronchoalveolar lavage AMs using a BD FACS Aria Fusion machine. One hundred thousand sorted cells were centrifuged at 500 × g for 10 min at 4 °C per replicate. Cells were lysed with lysis buffer (10 mM Tris-HCl pH 7.4, 10 mM MgCl$_2$, and 0.1% IGEPAL CA-630). Libraries were prepared using the TruePrep DNA library prep kit V2 for Illumina (Vazyme) according to the manufacturer's instructions. Libraries were cleaned up with AMPure XP beads (Beckman coulter) at a ratio of 0.7 and the quality was assessed using the 2100 Bioanalyzer (Agilent Technologies). Libraries were sequenced with 150 paired ends using a NovaSeq 6000 instrument (Illumina) for an average of 20 million reads per sample.

## Analysis of bulk RNA-sequencing data
cDNA libraries were sequenced on an Illumina HiSeq X Ten platform and 150 bp paired-end reads were generated. Approximately 49.96 M raw reads were generated for each sample. Raw data (raw reads) in fastq format were initially processed using Trimmomatic and low-quality reads were removed to generate approximately 48.42 M clean reads for each sample for further analyses. The clean reads were mapped to the mouse genome (GRCm39) and hamster genome (BCM_Maur_2.0) using hisat2 version 2.2.1, then sorted using samtools version 1.15 for differentially expressed gene analysis. The raw count matrix was quantified using featureCounts version 2.0.1, and the transcripts per kilobase million (TPM) of each gene were calculated. Differential expression analysis was performed using R package DESeq2 version 1.34.0. A $P$ value < 0.05, and foldchange > 2 was set as the threshold for significantly different expressions. Hierarchical cluster analysis of differentially expressed genes (DEGs) was performed to determine the expression patterns in different groups and samples.

The enrichment analysis of DEGs through Gene Ontology (GO), Kyoto Encyclopedia of Genes and Genomes (KEGG) and gene-set enrichment analysis (GSEA) were performed using R package clusterprofile version 4.2.0 and fgsea version 1.20.0. Time-series analysis was performed using R package Mfuzz version 2.54.0. All visualizations related to RNA-seq analysis were made using R packages ggplot2 version 3.3.6, ComplexHeatmap version 2.10.0, and enrichplot version 1.14.1.

## Pre-processing and analysis of bulk ATAC-seq data
Quality control of the original ATAC-seq read file was performed using fasqc version 0.11.9 and multiqc version 1.12 software, and the raw data were trimmed using Trim_galore version 0.6.7 to remove the adaptors. The data were aligned to the GRCm39 and BCM_Maur_2.0 genome separately using Bowtie2 version 2.4.5 with the '--very-sensitive -X 2000' parameter, followed by sorting using samtools version 1.15. Duplicated and unpaired reads were removed using the picard 'MarkDuplicates' command. Reads with mapping quality <30, and reads aligned to the mitochondria chromosome were also removed. All downstream analyses were performed on the filtered reads. The bam file for all samples was converted to a bed file and then callpeak using MACS2 version 2.2.7.1 with the '-nomodel --shift -100 --extsize 200' parameter.

Differential peak analysis was processed using bedtools to merge the peak file and featureCounts version 2.0.1 was used to construct the matrix; DESeq2 was then used to identify the differential peaks.

Coverage files from filtered bam files were produced using deeptools version 3.5.1 bamCoverage command. Each position was

normalized with '−normalizeUsing RPGC,' followed by conversion to bigWig format and visualization using IGV software.

## Statistics and visualization

Statistical analyses were performed with Prism 8 (GraphPad software). Bars represent the mean. Statistical significance of two group was performed with Mann−Whitney tests. Statistical significance of three or more than three was performed with Kruskal−Wallis tests with Dunn's multiple comparisons test.

And the visualization of RNA-seq and scRNA-seq analysis was performed by using R package ggplot2 (version 3.3.6) and Complex-Heatmap (version 2.10.0).

## Reporting summary

Further information on research design is available in the Nature Portfolio Reporting Summary linked to this article.

## Data availability

The raw and processed data of RNA-seq, ATAC-seq and scRNA-seq experiment generated in this study have been deposited in the Gene Expression Omnibus (GEO) repository under SuperSeries accession code GSE227649. Any other raw data or non-commercial material used in this study are available from the corresponding author upon request. Source data are provided with this paper.

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

## Acknowledgements

This study was supported by National Natural Science Foundation of China grants 81991491 (to N.X.), 32170943 (to T.Z.), 31730029 (to N.X.), and 82041038 (to Yixin C.); Program on Key Research Project of China 2020YFC0842600 (to N.X.); Guangdong-Hongkong-Macau Joint Laboratory grant 2019B121205009 (to Y.G.); Natural Science Foundation of Fujian Province 2020J06007 (to T.Z.) and 2021J02006 (to Yixin C.); Xiamen Youth Innovation Fund Project 3502Z20206060 (to T.Z.); and Fundamental Research Funds for the Central Universities 20720220003 (to N.X.).

## Author contributions

Conceptualization, T.Z., and N.X.; Data curation, L.Z., Y.J., J.H., J.C., R.Q., L.Y., T.S., and Hui Z.; Formal analysis, L.Z., Y.J., and J.H.; Investigation, L.Z., Y.J., J.H., J.C., R.Q., L.Y., T.S., Hui Z., C.C., Yaode C., X.W., X.L., Q.G., C.Z., M.Z., J.M., W.L., M.Y., R.F., Yangtao W., F.C., H.X., M.N., Yiyi C., K.W., and M.F.; Methodology, L.Z., Y.J., J.H., J.C., R.Q., L.Y., and T.S.; Visualization, L.Z., Y.J., J.H., J.C., R.Q., L.Y., and T.S.; Validation, L.Z., Y.J., and R.Q.; Resources, L.Z., J.Z., Yixin C. T.Z., H.Q., Y.G., and N.X.; Funding acquisition, Yixin C., T.Z., Y.G., and N.X.; Project administration, T.Z.; Supervision, J.Z., T.W., Yixin C., T.Z., C.L., H.Q., Y.G., and N.X.; Writing-original draft, L.Z., Y.J., J.H., and T.Z.; Writing-review & editing, Yingbin W., Z.Z., S.H., S.G., S.C., Huachen Z., T.C., Q.Y., T.W., J.Z., Yixin C., T.Z., C.L., H.Q., Y.G., and N.X.

## Competing interests
The authors declare no competing interest.

## Additional information

[1]State Key Laboratory of Molecular Vaccinology and Molecular Diagnostics; National Institute of Diagnostics and Vaccine Development in Infectious Diseases, School of Public Health & School of Life Sciences, Xiamen University, 361102 Xiamen, Fujian, China. [2]Tsinghua-Peking Center for Life Sciences, Laboratory of Dynamic Immunobiology, School of Medicine, Tsinghua University, 100084 Beijing, China. [3]National Institute for Food and Drug Control, 102629 Beijing, China. [4]State Key Laboratory of Cellular Stress Biology, School of Life Sciences, Xiamen University, 361102 Xiamen, Fujian, China. [5]Xiang An Biomedicine Laboratory, 361102 Xiamen, Fujian, China. [6]State Key Laboratory of Emerging Infectious Diseases, School of Public Health, Li Ka Shing Faculty of Medicine, The University of Hong Kong, Hong Kong 999077, China. [7]Guangdong-Hong Kong Joint Laboratory of Emerging Infectious Diseases/Joint Laboratory for International Collaboration in Virology and Emerging Infectious Diseases, Joint Institute of Virology (STU/HKU), Shantou University, 515063 Shantou, China. [8]These authors contributed equally: Liang Zhang, Yao Jiang, Jinhang He, Junyu Chen, Ruoyao Qi, Lunzhi Yuan, Tiange Shao, Hui Zhao. ✉e-mail: wuting@xmu.edu.cn; zhangj@xmu.edu.cn; yxchen2008@xmu.edu.cn; zhangtianying@xmu.edu.cn; changguili@aliyun.com; qihai@tsinghua.edu.cn; yguan@hku.hk; nsxia@xmu.edu.cn

