## [Peer Review File · Nature Communications]

Intranasal influenza-vectored COVID-19 vaccine restrains the SARS-CoV-2 inflammatory response in hamstersReviewers' comments:

Reviewer #1 (Remarks to the Author):

This is a study that investigates a very important and relevant biological problem: the possibility to develop a vaccine that could protect against multiple SARS-CoV-2 variants by inducing both T-cell immunity and innate/trained immunity. The authors use a NS1-deleted H1N1 vector carrying the gene encoding SARS-CoV-2-RBD to build the candidate vaccine, and perform a series of immunological studies by intra-nasal administration. The experiments are well-designed, but there are a number of clarifications that are needed.

Comments:

1. One of the major concerns is that the control group in many experiments consisted in PBS administration, rather than the 'empty' NS1-deleted H1N1 vector. For example, in Figures 1,5,6 a PBS control is used, while in others there is also the NS1-deleted H1N1 vector. Why this discrepancy?
2. A second major concern relates to the number of animals used in some experiments, which were as low as 2-4 for some of the figures shown. This number is far too low to be able to be confident on the strength of the results.
3. Also, reproducibility of the data in independent experiments is needed to be sure of the conclusions.
4. In Figure 2, the authors studied IFN γ production capacity. This is important, but what was also the production of type I interferons after vaccination, as they are known very important anti-viral mechanisms.
5. In order to characterize more robustly the trained immunity phenotype, production of cytokines by AMs would be useful as well. Similarly, the epigenetic analysis suggests changes in glycolysis: was lactate production higher after vaccination?
6. Can the authors speculate whether the protection would be induced only against a respiratory infection, or could it be envisioned to have also systemic effects?

Reviewer #2 (Remarks to the Author):

This is a comprehensive piece of work that addresses the immune consequences of intranasal delivery of NS1-deleted nasal vaccines in mice and hamsters. The vector is compared with or without a pre-Omicron RBD domain.

I note that the vector is currently in clinical trial. Phase 1/II data showed that systemic T cell and antibody responses could be induced in a minority of subjects.

The manuscript addresses many issues and as such the main points are somewhat unclear.

It is not unexpected that local delivery of a viral vector will induce an innate and trained immune response. Indeed, in the short term this would be expected to provide some clinical protection against SARS-CoV-2 challenge. The interferon/innate response is not virus-specific. As such I would suggest that this is covered much more in the interpretation and discussion of the manuscript.

As such the main interest in the paper is insight into local immune responses that are delivered additively with the RBD domain. As I understand it, protection studies were undertaken only after RBD delivery which may be an ethical constraint

To the reader there appears to be too much information in the manuscript which could probably be reduced by 50%

Figure 1 is in a limited murine dataset of 3 mice in each set and shows that viral vectors alter local immune profile over the short term. Interest here is limited.

Figure 2 if more valuable. What are virtual memory T cells much more boosted by the RBD vaccine? It was unclear what figure 1 was as this states 'protected'. The statistical analysis of Figure 3 could be brought out - otherwise it is a series of datasets. Figure 4 D and E are the most interesting and perhaps the focus should be in this area. I would like to see a clearer assessment of how innate immunity is boosted with RBD domain (and if so, why) and then bring the novelty (if present) of the adaptive immune response to the RBD in the respiratory tissue.

Reviewer #3 (Remarks to the Author):

General comment.

The authors have previously shown through studies in animal models that intranasal dNS1-RBD vaccination can induce protection against a wide range of mutant strains of infection, including beta and omicron strains, without induction of neutralizing antibodies (Chen, J. et al. *Sci Bull*, 2022). Furthermore, this vaccine is already in clinical trials, and it has been reported that it can induce a SARS-CoV-2-specific cellular immune response in 44% of individuals one month after the second vaccination (Zhu, F. et al. *Lancet Respir Med*, 2022). However, the immunological mechanism of protection by this vaccine remained unclear. In this study, the authors analyzed the protective immunity induced by dNS1-RBD using several animal models. The authors analyzed in detail the immune response in the local lung tissues after vaccination and claimed that the induction of local cellular immunity is an important mechanism of action of this vaccine. The vaccine is currently undergoing a Ph3 trial, and its efficacy in humans is expected to become obvious through that study. However, studies using animal models, such as the present study, are essential to discuss the immunological mechanism of the protective effect of the vaccine in humans. In this regard, the research objective of this manuscript is considered important and of high scientific value. However, the manuscript is not convincing due to some immaturity in the study design and the experiments.

The vaccine is remarkable in that it induces a protective effect against viral infection in animal models despite a poor antibody response (Chen, J. et al. *Sci Bull*, 2022). Therefore, quantitative evaluation of the immunity induced by the vaccine is essential to prove that the protective effect observed in animal models can be obtained in humans. In general, it is very difficult to distinguish between an immune response and a pathological response to live vaccines since the phenomenon of infection of the inoculated virus is essential for inducing immunity. It is considered that the higher the dose of virus inoculation, the higher the efficiency of virus replication after infection and the higher the immune response, but on the other hand, the stronger the pathological response. Therefore, in live vaccine development, dose setting is considered an extremely important factor in determining vaccine efficacy and safety. In particular, the susceptibility of vaccine viruses varies from host to host, making it difficult to determine the dose based on the results in animal models, as is the case with inactivated vaccines, to predict the amount of immune response in humans based on body weight and body surface area. In mouse models, animals have received one dose of this live vaccine at 5×10^4 PFU, and hamsters have received two doses of this vaccine at 1×10^5 PFU (Chen, J. et al. *Sci Bull*, 2022). Humans, on the other hand, have received two doses of 1×10^6 PFU vaccine in clinical trials (Zhu, F. et al. *Lancet Respir Med*, 2022). Although the vaccine doses are distinct for each host, the paper lacks data to discuss the extrapolation of the immune responses observed in the animal models to humans, making it significantly difficult to evaluate the significance of this manuscript.

In this regard, the data presented in this manuscript are too immature to support the authors' claims. In conclusion, I suggest that quite a lot of improvements are needed to publish this manuscript in the *Nature Communications*.

Major comment 1.

To induce broad-spectrum cellular immunity to various variants, a vaccine using conserved

sequences rather than RBDs with diverse sequences would be advantageous, but the reasons why the authors used RBDs were not stated. To understand the immunological mechanism of the defense mechanism of this vaccine, comparative experiments with vaccines using SARS-CoV-2 sequences other than RBD seem to be appropriate. In particular, the RBD region is considered important for the induction of neutralizing antibodies but is not considered a promising region for the induction of cellular immunity.

Major comment 2.

In the dNS1-RBD vaccine, the RBD is trimerized by foldon, but no data confirming the extracellular expression of the trimerized RBD by vaccination have been found either in a previous report (Chen, J. et al. *Sci Bull*, 2022) or in this manuscript. Considering the weak neutralizing antibody induction in this vaccine, the expression of RBDs inserted into the genome of influenza viruses may be weak but obtaining data comparing the expression of the spike protein with that of mRNA vaccines would be useful in understanding the mechanism of immune response to this vaccine.

Major comment 3.

The dNS1-RBD vaccine has the genome of influenza virus and should also induce an immune response against influenza virus, and comparative analysis of the immune response against SARS-CoV-2 and that against influenza virus would be useful to understand the mechanism of the immune response of this vaccine. The efficacy of this vaccine in the presence of pre-existing immunity to influenza should also be evaluated.

Reviewers' comments:

Reviewer #1 (Remarks to the Author):

This is a study that investigates a very important and relevant biological problem: the possibility to develop a vaccine that could protect against multiple SARS-CoV-2 variants by inducing both T-cell immunity and innate/trained immunity. The authors use a NS1-deleted H1N1 vector carrying the gene encoding SARS-CoV-2-RBD to build the candidate vaccine, and perform a series of immunological studies by intra-nasal administration. The experiments are well-designed, but there are a number of clarifications that are needed.

Response: Thank you for your overall positive comments and insightful suggestions that allowed us to improve the quality of the manuscript. We have great news to share with you. Encouraging efficacy and safety data of this vaccine has been obtained from a multi-centre randomized, double-blind, placebo-controlled phase 3 clinical trial during Omicron period, and has been granted emergency use authorization in the Chinese mainland. This candidate intranasal spray influenza virus-vectored COVID-19 vaccine conferred broad-spectrum protection against symptomatic COVID-19 caused by SARS-CoV-2 Omicron variant in adults regardless of age or underlying medical conditions, both as the primary schedule or heterologous booster doses. Efficacy data showed that dNS1-RBD conferred 100% (95% CI, -9.2 to 100.0, log-rank $P=0.0255$) efficacy against hospitalized COVID-19. For the endpoint of COVID-19 with 3 or more related symptoms ($OSI \geq 3$), vaccine efficacy within three months after the second dose was 66.7% (95% CI, 8.3 to 87.9) in the naïve cohort, and for participants who had previously received inactivated vaccine, the estimated relative vaccine efficacy within 6 months after the second dose was 63.1% (95% CI -15.8 to 88.3). For all the symptomatic COVID-19, the above two efficacies were 55.2% (95% CI 13.8 to 76.7) and 46.5% (95% CI -26.3 to 77.3), respectively. Furthermore, the estimated vaccine efficacy did not differ across age groups (18-59 years and ≥ 60 years of age) and underlying medical conditions, as with the safety results.

Comments:

1. One of the major concerns is that the control group in many experiments consisted in PBS administration, rather than the 'empty' NS1-deleted H1N1 vector. For example, in Figures 1,5,6 a PBS control is used, while in others there is also the NS1-deleted H1N1 vector. Why this discrepancy?

Responses:

Thank you very much for your comment. Initially, the PBS group was used as a control to evaluate the efficacy of dNS1-RBD and to observe the transcriptome expression of the animal under the normal conditions. With the progresses of the research, we need to verify the effects of the NS1-deleted H1N1 vector. Interestingly, this vector had the similar immune responses as dNS1-RBD in some respects and also provides some protective effects against SARS-CoV-2. To further study, we performed sc-RNA-seq to observe the transcriptional signature of the vector and CA04, and the data were integrated in our revised manuscript.

2. A second major concern relates to the number of animals used in some experiments, which

were as low as 2-4 for some of the figures shown. This number is far too low to be able to be confident on the strength of the results.

Responses:

Thank you very much for your comment. In some transcriptome sequencing studies, authors used to set 3-6 replicates to increase confidence. In this study, we had set 3 replicates RNA-seq in mouse model, flow cytometry and ATAC-seq and sc-RNA-seq were performed to verify the consequence of RNA-seq. As for the hamster, we had set half male and half female to reduce transcriptome differences caused by gender after challenged by SARS-CoV-2, however, due to the limited resources of biosafety level 3 laboratory, we had to set 4 hamsters at each time point.

3. Also, reproducibility of the data in independent experiments is needed to be sure of the conclusions.

Responses:

Thank you very much for your comment. In terms of vaccine efficacy, we conducted challenge experiments at multiple time points after vaccine immunization, all of which showed stable protective effects, which was also approved by phase III clinical trial. In terms of immune response data, the conclusions we obtained through ELISPOT, flow cytometry, and single-cell transcriptome sequencing (in revised manuscript) are consistent with each other.

4. In Figure 2, the authors studied IFN γ production capacity. This is important, but what was also the production of type I interferons after vaccination, as they are known very important anti-viral mechanisms.

Responses:

Thank you very much for your valuable comment. After immunization, in addition to IFN γ production, type I interferon responses with antiviral effects can also be induced. The upregulation of type I interferons was not only detected by single cell RNA-seq in our revised manuscript and also by Luminex assay to detect the protein level of cytokines in the lung tissue (Fig 5. Chen, J. et al. Sci Bull, 2022). As expected, the upregulation of IFN α in dNS1-RBD and dNS1-Vector group was much faster and higher than that in wild type influenza virus (CA04) group, indicating that the deletion of NS1 relieves the antagonism of the influenza virus on the type I interferon pathway and can rapidly induce an antiviral innate immune response.

Fig 5. Chen, J. et al. Sci Bull, 2022

5. In order to characterize more robustly the trained immunity phenotype, production of cytokines by AMs would be useful as well. Similarly, the epigenetic analysis suggests changes in glycolysis: was lactate production higher after vaccination?

Responses:

Thank you very much for the comment. In the early post-vaccination period, we found that the levels of cytokines gradually decreased over time in the whole lung tissue, showing a return to resting state levels. The same conclusion was also found in sc-RNA-seq. The cytokine-related genes of AMs were up-regulated at the transcriptome level, but gradually returned to a resting state as time increased. But in the ATAC, the chromatin of some cytokines and antiviral effects was still open even after immunization two months, showing the potential to respond to the virus. And the upregulation of glycolytic pathway was also observed.

6. Can the authors speculate whether the protection would be induced only against a respiratory infection, or could it be envisioned to have also systemic effects?

Responses: Thank you for your interesting question. The results showed that the level of cellular immune response induced by this intranasal vaccine in the local respiratory tract is dozens of times higher than that in peripheral blood, and the effects of training immunity also mainly occur locally, so we speculate that the protective effect of the intranasal spray vaccine on respiratory tract infection would be much better than that against systemic infection. Nevertheless, partial specific T cell responses can be detected in peripheral blood in clinical trials and animal experiments, it should also confer some effects against systemic infection, but it will be relatively weaker than the local protective effect.

Reviewer #2 (Remarks to the Author):

This is a comprehensive piece of work that addresses the immune consequences of intranasal delivery of NS1-deleted nasal vaccines in mice and hamsters. The vector is compared with or without a pre-Omicron RBD domain.

I note that the vector is currently in clinical trial. Phase 1/II data showed that systemic T cell and antibody responses could be induced in a minority of subjects.

The manuscript addresses many issues and as such the main points are somewhat unclear. It is not unexpected that local delivery of a viral vector will induce an innate and trained immune response. Indeed, in the short term this would be expected to provide some clinical protection against SARS-CoV-2 challenge. The interferon/innate response is not virus-specific. As such I would suggest that this is covered much more in the interpretation and discussion of the manuscript.

As such the main interest in the paper is insight into local immune responses that are delivered additively with the RBD domain. As I understand it, protection studies were undertaken only after RBD delivery which may be an ethical constraint.

To the reader there appears to be too much information in the manuscript which could probably be reduced by 50%.

Responses: Thank you for your insightful comments and valuable suggestions that allowed us to improve the quality of the manuscript. We rearranged the manuscript following your suggestion and added a series of dynamic single-cell RNA sequencing data. These new results further support the major conclusions and increase the reliability of our study. We hope that your comments have been addressed accurately. We have revised the manuscript in accordance with the comments and marked all the amendments on our revised manuscript.

We have great news to share with you. Encouraging efficacy and safety data of this vaccine has been obtained from a multi-centre randomized, double-blind, placebo-controlled phase 3 clinical trial during Omicron period, and has been granted emergency use authorization in the

Chinese mainland. This candidate intranasal spray influenza virus-vectored COVID-19 vaccine conferred broad-spectrum protection against symptomatic COVID-19 caused by SARS-CoV-2 Omicron variant in adults regardless of age or underlying medical conditions, both as the primary schedule or heterologous booster doses. Efficacy data showed that dNS1-RBD conferred 100% (95% CI, -9.2 to 100.0, log-rank $P=0.0255$) efficacy against hospitalized COVID-19. For the endpoint of COVID-19 with 3 or more related symptoms ($OSI \geq 3$), vaccine efficacy within three months after the second dose was 66.7% (95% CI, 8.3 to 87.9) in the naïve cohort, and for participants who had previously received inactivated vaccine, the estimated relative vaccine efficacy within 6 months after the second dose was 63.1% (95% CI -15.8 to 88.3). For all the symptomatic COVID-19, the above two efficacies were 55.2% (95% CI 13.8 to 76.7) and 46.5% (95% CI -26.3 to 77.3), respectively. Furthermore, the estimated vaccine efficacy did not differ across age groups (18-59 years and ≥ 60 years of age) and underlying medical conditions, as with the safety results.

Figure 1 is in a limited murine dataset of 3 mice in each set and show that viral vectors alter local immune profile over the short term. Interest here is limited.

Responses: Thank you very much for your valuable comment. We performed scRNA-seq at multiple time points, a higher resolution of pulmonary immune response elicited by dNS1-RBD, dNS1-Vector, and CA04 was obtained (revised Figure 1). Consistent with our previous bulk RNA-seq data, both dNS1-RBD and dNS1-Vector vaccinated mice exhibited a rapid, strong, and "transient" innate immune response distinctly from its parental CA04 virus-infected mice, which displayed an untimely innate immune response and cause persistent and severe lung inflammation (revised Figure 2 and Figure 4). At 44 days post-last immunization, as compared with uninfected mice lungs, we observed an increase in both the proportion and number of non-classical monocytes (30% vs. 17%), which play important role in patrolling, sensing viral nucleic acids, and initiating the innate immune response. A higher percentage and number of NKG2C+ NK cells (75% vs. 66%) were annotated. This cluster of cells is supposed to unleash NK cell anti-viral activity in SARS-CoV-2 infection and possesses a memory-like feature that is antigen-unspecific. In addition, at 44 dpi., apart from increases in non-classical monocytes and NK cells, expansion of T cells including CD4+ and CD8+ T cells was observed. Tissue-resident memory T cells (TRM) were induced in both the upper (URT) and lower respiratory tract (LRT). The URT is one of the first contact sites for inhaled pathogens and immune systems in humans. In mice, nasal lymphoid tissues (NALT) are thought to be the functional equivalent of the human tonsil. The TRMs generated in the NALT could effectively block the virus transmission from the URT to the lung.

Figure 2 if more valuable. What are virtual memory T cells much more boosted by the RBD vaccine? It was unclear what figure I was as this states 'protected'.

Responses: Thank you very much for the comment. This data has been removed in the revised manuscript

The statistical analysis of Figure 3 could be brought out - otherwise it is a series of datasets.

Responses: Thank you very much for the comment. In this section, assay of transposase accessible chromatin sequencing (ATAC-seq) analysis was used to identify the potential

changes that may occur in the chromatin accessibility. Integrative Genomics Viewer (IGV) was used to visualize the peaks of ATAC-seq. However, the statistical analysis of peaks is still limited using this software, and this is a common limitation needed to be addressed in this area.

Figure 4 D and E are the most interesting and perhaps the focus should be in this area

Responses: Thank you very much for the comment. We have rearranged the manuscript follow your suggestion and focused more about the local cellular responses and profile of cell differentiation.

Reviewer #3 (Remarks to the Author):

General comment.

The authors have previously shown through studies in animal models that intranasal dNS1-RBD vaccination can induce protection against a wide range of mutant strains of infection, including beta and omicron strains, without induction of neutralizing antibodies (Chen, J. et al. *Sci Bull*, 2022). Furthermore, this vaccine is already in clinical trials, and it has been reported that it can induce a SARS-CoV-2-specific cellular immune response in 44% of individuals one month after the second vaccination (Zhu, F. et al. *Lancet Respir Med*, 2022). However, the immunological mechanism of protection by this vaccine remained unclear. In this study, the authors analyzed the protective immunity induced by dNS1-RBD using several animal models. The authors analyzed in detail the immune response in the local lung tissues after vaccination and claimed that the induction of local cellular immunity is an important mechanism of action of this vaccine. The vaccine is currently undergoing a Ph3 trial, and its efficacy in humans is expected to become obvious through that study. However, studies using animal models, such as the present study, are essential to discuss the immunological mechanism of the protective effect of the vaccine in humans. In this regard, the research objective of this manuscript is considered important and of high scientific value. However, the manuscript is not convincing due to some immaturity in the study design and the experiments.

The vaccine is remarkable in that it induces a protective effect against viral infection in animal models despite a poor antibody response (Chen, J. et al. *Sci Bull*, 2022). Therefore, quantitative evaluation of the immunity induced by the vaccine is essential to prove that the protective effect observed in animal models can be obtained in humans. In general, it is very difficult to distinguish between an immune response and a pathological response to live vaccines since the phenomenon of infection of the inoculated virus is essential for inducing immunity. It is considered that the higher the dose of virus inoculation, the higher the efficiency of virus replication after infection and the higher the immune response, but on the other hand, the stronger the pathological response. Therefore, in live vaccine development, dose setting is considered an extremely important factor in determining vaccine efficacy and safety. In particular, the susceptibility of vaccine viruses varies from host to host, making it difficult to determine the dose based on the results in animal models, as is the case with inactivated vaccines, to predict the amount of immune response in humans based on body weight and body surface area. In mouse models, animals have received one dose of this live vaccine at 5×10^4 PFU, and hamsters have received two doses of this vaccine at 1×10^5 PFU (Chen, J. et al. *Sci Bull*, 2022). Humans, on the other hand, have received two doses of 1×10^6 PFU vaccine in

clinical trials (Zhu, F. et al. Lancet Respir Med, 2022). Although the vaccine doses are distinct for each host, the paper lacks data to discuss the extrapolation of the immune responses observed in the animal models to humans, making it significantly difficult to evaluate the significance of this manuscript.

In this regard, the data presented in this manuscript are too immature to support the authors' claims. In conclusion, I suggest that quite a lot of improvements are needed to publish this manuscript in the Nature Communications.

Response: Thanks for your insightful comments and valuable comments on our manuscript. You have pointed out the critical challenges for development of live-attenuated vaccines. The lack of quantitatively immune markers is the limitation of this vaccine. Although the contribution of cellular immunity on the COVID-19 protection was well recognized, the correlation of the T cell responses level and protection effects are still not established, and the trained immunity is even lacking quantification assay. The lack of quantitative immune indicators associated with protection is the common limitation in the field. However, intranasal vaccine-induced local tissue-resident T cells and trained immunity located in the respiratory tract can cope with the accelerated emergence of various mutant strains that escaping neutralizing antibodies.

In the absence of “correlates of protection” indicator, phase III clinical efficacy trials are the only way to verify vaccine protection. We have great news to share with you. Encouraging efficacy and safety data of this vaccine has been obtained from a multi-centre randomized, double-blind, placebo-controlled phase 3 clinical trial during Omicron period, and has been granted emergency use authorization in the Chinese mainland. This candidate intranasal spray influenza virus-vectored COVID-19 vaccine conferred broad-spectrum protection against symptomatic COVID-19 caused by SARS-CoV-2 Omicron variant in adults regardless of age or underlying medical conditions, both as the primary schedule or heterologous booster doses. Efficacy data showed that dNS1-RBD conferred 100% (95% CI, -9.2 to 100.0, log-rank $P=0.0255$) efficacy against hospitalized COVID-19. For the endpoint of COVID-19 with 3 or more related symptoms ($OSI \geq 3$), vaccine efficacy within three months after the second dose was 66.7% (95% CI, 8.3 to 87.9) in the naïve cohort, and for participants who had previously received inactivated vaccine, the estimated relative vaccine efficacy within 6 months after the second dose was 63.1% (95% CI -15.8 to 88.3). For all the symptomatic COVID-19, the above two efficacies were 55.2% (95% CI 13.8 to 76.7) and 46.5% (95% CI -26.3 to 77.3), respectively. Furthermore, the estimated vaccine efficacy did not differ across age groups (18-59 years and ≥ 60 years of age) and underlying medical conditions, as with the safety results.

We agree that the current immunological knowledge and the evidence provided in this manuscript was not enough to fully explain the broad-spectrum protective effects that approved in phase III clinical trials, we hoped that the staged progress shown in this study is still worth publishing because of the number of SARS-CoV-2 variants resistance to neutralizing antibodies continuously emerging and remain a major threat to global health, antibody-independent COVID-19 protection represents a unique advantage of the intranasal vaccine dNS1-RBD against emerging variants. This research may attract more scientists to pay attention to and further study mucosal vaccines and promote the development of more effective strategies against COVID-19.

In addition, in our revised manuscript, we further described the differentiation landscape of lymphocytes and myeloid cells after vaccination through single-cell transcriptome sequencing and found that compared with wild-type influenza viruses as well as mock control mice, vaccine virus strains and vector virus strains with NS1 gene deletion can induce a higher proportion of cell subpopulations with anti-inflammatory functions.

Major comment 1.

To induce broad-spectrum cellular immunity to various variants, a vaccine using conserved sequences rather than RBDs with diverse sequences would be advantageous, but the reasons why the authors used RBDs were not stated. To understand the immunological mechanism of the defense mechanism of this vaccine, comparative experiments with vaccines using SARS-CoV-2 sequences other than RBD seem to be appropriate. In particular, the RBD region is considered important for the induction of neutralizing antibodies but is not considered a promising region for the induction of cellular immunity.

Responses: Thank you for your comments, we very much agree. The actual situation is that at the beginning of the epidemic, there was no enough knowledge about SARS-CoV-2. Spike was considered to be the most important vaccine target, but the loading capacity of the influenza vector is not large enough to carry the full length of spike. Therefore, RBD was selected for construction of vaccine strain. We agree with you and believe that there has better selections to induce stronger cellular immunity. However, since this vaccine strain was produced during the emergency stage at the beginning of the COVID-19 pandemic, and phase III clinical trials have been completed, the purpose of our study is to describe the protective immune responses induced by this vaccine, the construction of vaccine strain was not the focus of this study. We will consider and follow your suggestion in the future study.

Major comment 2.

In the dNS1-RBD vaccine, the RBD is trimerized by foldon, but no data confirming the extracellular expression of the trimerized RBD by vaccination have been found either in a previous report (Chen, J. et al. *Sci Bull*, 2022) or in this manuscript. Considering the weak neutralizing antibody induction in this vaccine, the expression of RBDs inserted into the genome of influenza viruses may be weak but obtaining data comparing the expression of the spike protein with that of mRNA vaccines would be useful in understanding the mechanism of immune response to this vaccine.

Responses: Thank you for your meticulous review of the paper. The expression of RBD was visualized using confocal analysis and further confirmed by Western blot, the trimeric form of RBD was detected by western blot (Fig 1D. Chen, J. et al. *Sci Bull*, 2022). The relative lower expression level compared to mRNA vaccines might be a reason of weak antibody response.

Fig 1D. Chen, J. et al. Sci Bull, 2022

Major comment 3.

The dNS1-RBD vaccine has the genome of influenza virus and should also induce an immune response against influenza virus, and comparative analysis of the immune response against SARS-CoV-2 and that against influenza virus would be useful to understand the mechanism of the immune response of this vaccine. The efficacy of this vaccine in the presence of pre-existing immunity to influenza should also be evaluated.

Responses: Thank you for the comment. In the preclinical study, we have evaluated the immune response against influenza virus. The results showed that NP-specific IgG and Flu-specific cellular immune response was detected at 14 days after boost immunization with dNS1-RBD and CA04-dNS1 virus. Meanwhile, the neutralizing antibodies against H5N1 was undetectable after dNS1-RBD immunization, but the protection effects against homologous and heterologous influenza virus challenges (H1N1 and H5N1) were observed, which demonstrating that dNS1-RBD provide better cross-protective activity than CA04-WT virus (Chen, J. et al. Sci Bull, 2022). In another study cited below “Generation of DelNS1 Influenza Viruses: a Strategy for Optimizing Live Attenuated Influenza Vaccines. mBio, 17 Sep 2019, 10(5)”, Wang also proved that CA4-DELNS1 virus can provide better protection against H5N1 than CA4-LAIV (cold-adapted virus). All above results suggested that dNS1-RBD could induce H1N1 specific antibodies and NP-specific T cell response, while provide broad protection against influenza a virus. The trained immunity against influenza virus induced by dNS1-RBD might also not be neglected.

Wang et al., mBio, 17 Sep 2019, 10 (5)

To further explore the influences of pre-existing H1N1 antibody on the immune response induced by our intranasal spray vaccine, Coalition for Epidemic Preparedness Innovations (CEPI) commissioned Public Health England (PHE) in early 2020 to conduct an experimental study on our intranasal vaccine in ferrets, an influenza virus ultra-sensitive animal model. As shown in the figure below, the result suggested that prior influenza vaccination did not attenuate the immune response induced by the intranasal spray vaccine (QLAIV, a quadrivalent live attenuated influenza vaccine).

RBD-binding IgG antibody (ELISA)

It is noteworthy, results in animals may differ from those in humans. We will evaluate whether pre-existing immunity affects the efficacy of this nasal spray in the real world.

REVIEWER COMMENTS

Reviewer #1 (Remarks to the Author):

The authors responded appropriately to my suggestions.

Reviewer #2 (Remarks to the Author):

The authors have made a number of amendments to the manuscript which help to build confidence in the analysis.

It is noteworthy and important that the vaccine is now in clinical deployment

Reviewer #3 (Remarks to the Author):

Comment 1:

The vaccine studied in this manuscript is already in the final stages of clinical trials, and the preclinical studies at this point required are not only proof of concept, but also the scientific rationale for properly evaluating the performance of this vaccine in humans. In this final phase of vaccine development, it is critical to elucidate the quantitative indexes of immunity correlated of protection (CoP), as well as to understand the nature of immunity that contributes to vaccine efficacy. For vaccines based on novel modalities with distinct mechanism of actions (MoAs) from current vaccines, it is necessary to clarify the CoP to understand the transition of the real-world efficacy of the vaccine after their clinical implementation. Because SARS-CoV-2 evades humoral immunity as mutations accumulate, understanding the quality and quantity of immunity that contributes to the efficacy of this vaccine should provide useful information for estimating the vaccine's ability to adapt to changes in SARS-CoV-2 epidemic strains once the vaccine is available for clinical use. However, the authors mentioned that there are no quantitative immune surrogates to assess the efficacy of this vaccine in the response letter of this revised manuscript.

Although it has been suggested that the MoA of this vaccine is to induce both T-cell immunity and innate/trained immunity, no quantitative assessment has been conducted in this paper to discuss the different degrees of contribution of the two types of immunity to the vaccine efficacy. As the authors note, the partial protective effect against SARS-CoV-2 infection even in dNS1_Vector without RBDs (Fig. 6F) is an important data showing that both innate/trained immunity and T-cell immunity are independently responsible for the MoA of this vaccine. Furthermore, the results indicate that at least T-cell immunity plays an important role in protecting against SARS-CoV-2 infection. As the authors recognize, the correlation between T-cell response levels and vaccine efficacy has not yet been established in human SARS-CoV-2 infection, and even quantitative assays for trained immunity are lacking. However, it is possible to obtain quantitative data on T-cell response levels, both in humans and in animal models. Preclinical studies with animal models can provide a valuable opportunity to assess the relationship between the amount of immunity induced by this vaccine and vaccine efficacy against SARS-CoV-2 infection and understand the dose-response of the immunity induced by this vaccine to preventing infection. It is extremely difficult to conduct experiments with such objectives on human subjects. Then, it is an important role of preclinical animal studies to provide a scientific rationale addressing these questions.

Specifically, it would be possible to quantitatively evaluate the degree of contribution of the T-cell response level to protection against infection by comparing the protection against infection among animal groups with distinct T-cell response levels prepared by varying the vaccine dose to the animals. In addition, only the rate of weight loss was evaluated in the infection experiments compared to dNS1_Vector, and the evaluation of viral titer was lacking. To understand the quantitative differences in immunity induced by dNS1_Vector and

dNS1_RBD, the suppression of viral infection is essential. It would also be extremely useful in quantitatively understanding the immunity related to the MoA of this vaccine. Furthermore, extrapolation of the induction of immunity by live vaccines in rodent animal models to humans has significant limitations. However, by obtaining the above data and understanding the MoA of this vaccine quantitatively, it should be possible to estimate the MoA of this vaccine in humans by compensating for differences in induced immunity due to differences in each animal species' susceptibility to this vaccine virus. Therefore, additional SARS-CoV-2 challenge experiments to reveal the dose-response of the immunity induced by this vaccine to prevent infection would overcome the major limitation in this manuscript and encourage the successful development of this vaccine.

Comment 2:

Does natural or acquired immunity affect the nature of T-cell immunity induced by this vaccine against SARS-CoV-2? Using the existing samples from the hamster SARS-CoV-2 challenge models shown in this manuscript, it would be useful to evaluate the difference between the nature of T-cell immunity induced by natural infection with SARS-CoV-2 and that by this vaccine.

Reviewer comments and point-by-point response

Reviewer #1 (Remarks to the Author):

The authors responded appropriately to my suggestions.

Responses:

Thank you for your feedback. We're glad to hear that our responses were appropriate and helped improve the quality of the manuscript. Your insights have been incredibly valuable to us.

Reviewer #2 (Remarks to the Author):

The authors have made a number of amendments to the manuscript which help to build confidence in the analysis.

It is noteworthy and important that the vaccine is now in clinical deployment.

Responses:

Thank you for your feedback. We're glad to hear that the amendments we made have helped build confidence in our analysis. We're also thrilled about the vaccine's clinical deployment, and this vaccine (Pneucolin) has been approved for emergency use in China on December 2, 2022. Your comments have been incredibly helpful, and we appreciate your time and effort in reviewing our manuscript.

Reviewer #3 (Remarks to the Author):

Comment 1:

The vaccine studied in this manuscript is already in the final stages of clinical trials, and the preclinical studies at this point required are not only proof of concept, but also the scientific rationale for properly evaluating the performance of this vaccine in humans. In this final phase of vaccine development, it is critical to elucidate the quantitative indexes of immunity correlated of protection (CoP), as well as to understand the nature of immunity that contributes to vaccine efficacy. For vaccines based on novel modalities with distinct mechanism of actions (MoAs) from current vaccines, it is necessary to clarify the CoP to understand the transition of the real-world efficacy of the vaccine after their clinical implementation. Because SARS-CoV-2 evades humoral immunity as mutations accumulate, understanding the quality and quantity of immunity that contributes to the efficacy of this vaccine should provide useful information for estimating the vaccine's ability to adapt to changes in SARS-CoV-2 epidemic strains once the vaccine is available for clinical use.

However, the authors mentioned that there are no quantitative immune surrogates to assess the efficacy of this vaccine in the response letter of this revised manuscript.

Although it has been suggested that the MoA of this vaccine is to induce both T-cell immunity and innate/trained immunity, no quantitative assessment has been conducted in this paper to discuss the different degrees of contribution of the two types of immunity to the vaccine efficacy. As the authors note, the partial protective effect against SARS-CoV-2 infection even in dNS1_Vector without RBDs (Fig. 6F) is an important data showing that both innate/trained immunity and T-cell immunity are independently responsible for the MoA of this vaccine. Furthermore, the results indicate that at least T-cell immunity plays an important role in protecting against SARS-CoV-2 infection. As the authors recognize, the correlation between T-cell response levels and vaccine efficacy has not yet been established in human SARS-CoV-2 infection, and even quantitative assays for trained immunity are lacking. However, it is possible to obtain quantitative data on T-cell response levels, both in humans and in animal models. Preclinical studies with animal models can provide a valuable opportunity to assess the relationship between the amount of immunity induced by this vaccine and vaccine efficacy against SARS-CoV-2 infection and understand the dose-response of the immunity induced by this vaccine to preventing infection. It is extremely difficult to conduct experiments with such objectives on human subjects. Then, it is an important role of preclinical animal studies to provide a scientific rationale addressing these questions.

Specifically, it would be possible to quantitatively evaluate the degree of contribution of the T-cell response level to protection against infection by comparing the protection against infection among animal groups with distinct T-cell response levels prepared by varying the vaccine dose to the animals. In addition, only the rate of weight loss was evaluated in the infection experiments compared to dNS1_Vector, and the evaluation of viral titer was lacking. To understand the quantitative differences in immunity induced by dNS1_Vector and dNS1_RBD, the suppression of viral infection is essential. It would also be extremely useful in quantitatively understanding the immunity related to the MoA of this vaccine.

Furthermore, extrapolation of the induction of immunity by live vaccines in rodent animal models to humans has significant limitations. However, by obtaining the above data and understanding the MoA of this vaccine quantitatively, it should be possible to estimate the MoA of this vaccine in humans by compensating for differences in induced immunity due to differences in each animal

species' susceptibility to this vaccine virus. Therefore, additional SARS-CoV-2 challenge experiments to reveal the dose-response of the immunity induced by this vaccine to prevent infection would overcome the major limitation in this manuscript and encourage the successful development of this vaccine.

Responses:

Thank you very much for these insightful comments and suggestions. We have conducted the experiment as you suggested and the results have been included in the revised manuscript. Your input has greatly improved the quality of our research, and we appreciate your time and effort in reviewing our manuscript.

We strongly agree that for respiratory mucosal vaccines there is a great need to define the Correlation of Protection (CoP) so that we can recognize and understand the MoA of this vaccine. Currently, only neutralizing antibodies are recognized as potential CoPs for COVID-19 vaccines, be considered as the CoP of SARS-CoV-2 vaccines^{1,2}, which represents a challenging scientific issue for vaccines administered via the respiratory route. We also fully agree with the comment that the protective effects shown in the dNS1-vector suggest that trained immunity and T cell immunity play protective roles independently in the SARS-CoV-2-infected hamster model. So, this will also be our main goal and task in the next stage. Based on your suggestion, we have supplemented Fig 2M, Fig 3D, Fig 5D and Fig S5, and the details are as follows:

With the dilution of the vaccine (10-fold gradient), both antigen-specific T cell and tissue-resident T cell responses were significantly decreased. The proportion of alveolar macrophages with the trained immunity phenotype also decreased significantly with dose reduction. Although the response level of T cell and trained immunity decreased sharply at a 1:10,000 dilution (1×10^2 PFU/mL) compared to the undiluted vaccine (1×10^6 PFU/mL), it was still a detectable positive level with a statistical difference compared to the unimmunized control group.

Fig 2M**Fig 3D****Fig 3D****Fig 4D**
(Fig 2M) Spots Forming Cells (SFC) per million cells of IFN- γ SFCs from lung were quantified after stimulation of a peptide pool covering the entire spike protein in control and vaccinated group (1×10⁶ PFU/mL, 1×10⁵ PFU/mL, 1×10⁴ PFU/mL, 1×10³ PFU/mL, 1×10² PFU/mL). n = 8 mice/group.

(Fig 3D) Bar graph depicting the frequency (First row) and the absolute number (Second row) of TRM in the lungs after dNS1-RBD immunization (1×10⁶ PFU/mL, 1×10⁵ PFU/mL, 1×10⁴ PFU/mL, 1×10³ PFU/mL, 1×10² PFU/mL). n=6 mice/group.

(Fig 4D) Statistical analysis plots for gMFI of MHC II on AMs in C57BL/6 mice after vaccine immunization (1×10⁶ PFU/mL, 1×10⁵ PFU/mL, 1×10⁴ PFU/mL, 1×10³ PFU/mL, 1×10² PFU/mL). n = 6 mice/group.

Data are presented as mean \pm SEM. Statistical analysis was two-way ANOVA with Bonferroni's multiple comparisons test. ns, non-significant. *p < 0.05, **p < 0.01, ***p < 0.001.

(A-B) IFN- γ ELISpot response images of the control group and dNS1-RBD group (1×10^6 PFU/mL, 1×10^5 PFU/mL, 1×10^4 PFU/mL, 1×10^3 PFU/mL, 1×10^2 PFU/mL).

Further, we performed SARS-CoV-2 challenge experiment to study the dose-effect relationship, in which, a total of 48 hamsters were divided into six groups, including five groups that receiving a 10-fold serial diluted doses of vaccine and one unimmunized control group. At day 7 post challenge, the average body weight changes of each group are +3.88% (1×10^6 PFU/mL), +1.41% (1×10^5 PFU/mL), -1.43% (1×10^3 PFU/mL), -2.5% (1×10^2 PFU/mL), and -13.98% (Control), respectively. Gross observations of lung tissues showed that the control group had a large number of bleeding areas and pathological foci of inflammatory damage, the high-dose group (1×10^5 PFU/mL and 1×10^6 PFU/mL) had no or only a small number of pathological injuries, and the low-dose group (1×10^2 PFU/mL and 1×10^3 PFU/mL) could observe more obvious bleeding points and pathological lesions, but still showed obvious protective effects, and these results were consistent with the trend of the cellular immune response. Regrettably, the hamsters in the 1:100 dilution group died unexpectedly due to IVC cage ventilation issues and thus, data for this dosage could not be obtained. The data of body weight change and lung tissue pathological damage showed that the high-dose groups exhibited better protective effects than the low-dose groups, and in which, the lowest-dose group that with detectable cellular immune responses still showed a protective effect compared to the control group.

These results indicated that dNS1-RBD protects hamsters against challenge with SARS-CoV-2 in a dose-dependent manner, and its protective effect is consistent with the trend of cellular immune response. However, it is still difficult to establish a quantitative correlation between cellular response level and protection in this stage, which is the challenge for the entire field, and we will follow your suggestions to further study this important issue in the future.

A

B

C

Fig. S5 Hamsters receiving varying doses of the vaccine demonstrated different protective effects against SARS-CoV-2 challenge.

(A) Schema of the experimental design.

(B) Body weight changes of hamsters after cohoused exposure were plotted. The average weight loss of each group at 7 dpi is indicated as a colored number.

(C) Gross lung images from dNS1-RBD vaccinated group (1×10^6 PFU/mL, 1×10^5 PFU/mL, 1×10^4 PFU/mL, 1×10^3 PFU/mL, 1×10^2 PFU/mL) and control groups.

Data are presented as mean \pm SEM. Statistical analysis was two-way ANOVA with Bonferroni's multiple comparisons test. ns, non-significant. * $p < 0.05$, ** $p < 0.01$, *** $p < 0.001$, **** $p < 0.0001$.

Comment 2:

Does natural or acquired immunity affect the nature of T-cell immunity induced by this vaccine against SARS-CoV-2? Using the existing samples from the hamster SARS-CoV-2 challenge models shown in this manuscript, it would be useful to evaluate the difference between the nature of T-cell immunity induced by natural infection with SARS-CoV-2 and that by this vaccine.

Responses:

We thank reviewer for this valuable and interesting comment. Unfortunately, since our Biosafety level 3 (BSL-3) lab is not yet equipped to evaluate T cell responses post SARS-CoV-2 challenge, we are unable to provide direct data comparing differences between the nature of T-cell immunity induced by natural infection with SARS-CoV-2 and that by this vaccine. In order to answer your concerns as much as possible, we added the predicted results of immune cell infiltration using the RNA-seq data. Vaccinated and control hamsters showed distinct temporal dynamics in immune cell response. Three and five days after infection with the SARS-CoV-2 virus, M1-type pro-inflammatory macrophages were significantly enriched in the lungs of the control group hamsters, consistent with previous literature reports, suggesting that the predicted results can reflect the actual immune infiltration to some extent. However, one day after infection, CD8+ T cells were significantly enriched in the immune cell infiltrates of the vaccinated group lung tissue, indicating that the T cell response generated by vaccination may differ from that produced by natural infection. It is hypothesized that the vaccinated group hamsters may have rapidly recruited antigen-nonspecific T cells to exert a non-specific antiviral effect. However, due to limitations in the

available bulk transcriptome data and bioinformatics analysis techniques, it was not feasible to perform a more detailed analysis of the specific functional subgroups of CD8+ T cells in this study. This is a limitation of the current study and a topic that warrants further investigation, and we will strive to address it in future research.

(A) Stacked barplot of the immune cell proportion predicted by ImmuCellAI in each sample.

(B) Chi-square test of the immune cell proportion predicted in each group, the point size is

proportional to the Pearson residual of the chi-square test, and the color represents the degree of association with the Pearson chi-square residual (red indicates positive correlation, blue indicates negative correlation).

(C) Bar graph depicting the predicted cell fraction of M1-like macrophage at 3dpi and 5dpi, and CD8+ T cell at 1 dpi.

- 1 Krammer, F. A correlate of protection for SARS-CoV-2 vaccines is urgently needed. *Nature Medicine* 27, 1147-1148, doi:10.1038/s41591-021-01432-4 (2021).
- 2 Koch, T., Mellinghoff, S. C., Shamsrizi, P., Addo, M. M. & Dahlke, C. Correlates of Vaccine-Induced Protection against SARS-CoV-2. *Vaccines* (Basel) 9, doi:10.3390/vaccines9030238 (2021).

REVIEWERS' COMMENTS

Reviewer #3 (Remarks to the Author):

I have carefully reviewed the authors' responses to my comments and I am pleased to see that the authors have addressed all of my concerns. This revisions have significantly improved the clarity and quality of the manuscript.

Reviewer comments and point-by-point response

Reviewer #3 (Remarks to the Author):

I have carefully reviewed the authors' responses to my comments and I am pleased to see that the authors have addressed all of my concerns. This revisions have significantly improved the clarity and quality of the manuscript.

Responses:

Thank you for your thorough review of our revision. We are pleased to receive your positive feedback and appreciate your acknowledgment of the revisions we made in response to your comments. It is gratifying to know that our efforts have resulted in an improved manuscript in terms of clarity and quality. Thank you once again for your valuable contribution to our work.